# A scalable platform to discover antimicrobials of ribosomal origin

Richard S. Ayikpoe[1,2,9], Chengyou Shi [2,3,9], Alexander J. Battiste [1,2,9], Sara M. Eslami[1,2], Sangeetha Ramesh [2,4], Max A. Simon [2,5], Ian R. Bothwell[1,2], Hyunji Lee[1,2], Andrew J. Rice [1,2], Hengqian Ren[2,3], Qiqi Tian[2,6], Lonnie A. Harris[1,2], Raymond Sarksian[1,2], Lingyang Zhu [7], Autumn M. Frerk[1,2], Timothy W. Precord[1,2], Wilfred A. van der Donk [1,2,5,6,8] ✉, Douglas A. Mitchell [1,2,4] ✉ & Huimin Zhao [1,2,3,5] ✉

Ribosomally synthesized and post-translationally modified peptides (RiPPs) are a promising source of new antimicrobials in the face of rising antibiotic resistance. Here, we report a scalable platform that combines high-throughput bioinformatics with automated biosynthetic gene cluster refactoring for rapid evaluation of uncharacterized gene clusters. As a proof of concept, 96 RiPP gene clusters that originate from diverse bacterial phyla involving 383 biosynthetic genes are refactored in a high-throughput manner using a biological foundry with a success rate of 86%. Heterologous expression of all successfully refactored gene clusters in *Escherichia coli* enables the discovery of 30 compounds covering six RiPP classes: lanthipeptides, lasso peptides, graspetides, glycocins, linear azol(in)e-containing peptides, and thioamides. A subset of the discovered lanthipeptides exhibit antibiotic activity, with one class II lanthipeptide showing low μM activity against *Klebsiella pneumoniae*, an ESKAPE pathogen. Overall, this work provides a robust platform for rapidly discovering RiPPs.

Ribosomally synthesized and post-translationally modified peptides (RiPPs) constitute a major group of natural products exhibiting a range of biological activities, including antibacterial and antiviral activities[1,2]. As antimicrobial resistance has become a growing crisis, RiPPs have garnered significant attention, with many possessing modes of action distinct from clinically used antimicrobials[3,4]. Of particular interest are RiPPs that are active against the ESKAPE pathogens[5–7], a group of six nosocomial pathogens that are capable of escaping biocidal activity of available antibiotics: *Enterococcus faecium, Staphylococcus aureus, Klebsiella pneumoniae, Acinetobacter baumannii, Pseudomonas aeruginosa, and Enterobacter* spp[8]. ESKAPE pathogens are recognized by the World Health Organization as critical or high priority for new antibiotic development[9]. RiPP biosynthetic gene clusters (BGCs) typically include genes encoding the precursor peptide, modification enzymes, and in some cases, a leader peptidase and transporters. After ribosomal synthesis of the linear precursor peptide, the N-terminal leader region of the peptide is recognized by the modification enzymes while the C-terminal core region of the peptide receives the post-translational modifications (PTMs) (Fig. 1). During further maturation, the leader region is removed and depending on the RiPP

[1]Department of Chemistry, University of Illinois at Urbana-Champaign, Urbana 61801 IL, USA. [2]Carl R. Woese Institute for Genomic Biology, University of Illinois at Urbana-Champaign, Urbana 61801 IL, USA. [3]Department of Chemical and Biomolecular Engineering, University of Illinois at Urbana-Champaign, Urbana 61801 IL, USA. [4]Department of Microbiology, University of Illinois at Urbana-Champaign, Urbana 61801 IL, USA. [5]Department of Bioengineering, University of Illinois at Urbana-Champaign, Urbana 61801 IL, USA. [6]Department of Biochemistry, University of Illinois at Urbana-Champaign, Urbana 61801 IL, USA. [7]School of Chemical Sciences NMR Laboratory, University of Illinois at Urbana-Champaign, Urbana 61801 IL, USA. [8]Howard Hughes Medical Institute, 4000 Jones Bridge Road, Chevy Chase 20815 MD, USA. [9]These authors contributed equally: Richard S. Ayikpoe, Chengyou Shi, Alexander J. Battiste. ✉ e-mail: vddonk@illinois.edu; douglasm@illinois.edu; zhao5@illinois.edu

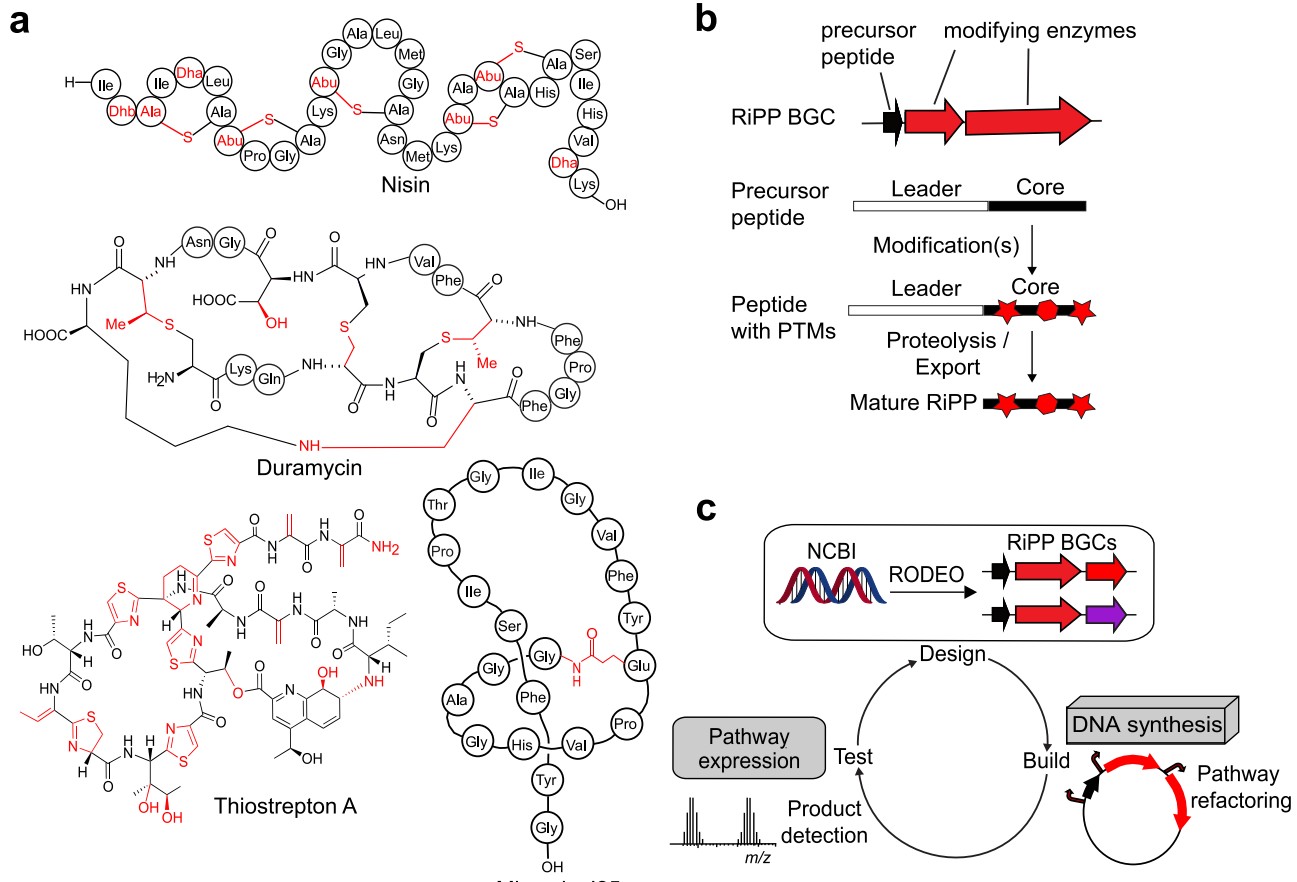

**Fig. 1 | Overview of RiPPs and FAST-RiPPs pipeline. a** Examples of RiPPs with biological activities relevant to human society. Nisin is used commercially in the food industry to combat food-borne pathogens[38], duramycin is used in diagnostics[76], thiostrepton is used in veterinary medicine[77], and microcin J25 is a bacterial RNA polymerase inhibitor[6]. Structural motifs formed during PTMs are shown in red. **b** Overview of RiPP biosynthesis. **c** Illustration of the FAST-RiPPs workflow.

class, additional PTMs are installed to yield the mature compound. Recent bioinformatic studies have revealed tens of thousands of prokaryotic RiPP BGCs that are predicted to encode new molecular scaffolds and potentially useful biological functions[10–19]. However, to determine which RiPPs possess antimicrobial activity, access to pure compound is desirable.

While initial RiPP discovery relied on conventional and serendipitous screening methods, the explosion of available genomic data and the advancement of genome-mining tools that enable the identification of RiPP BGCs in silico have facilitated the targeted discovery of RiPPs[20,21]. In particular, tools such as RODEO (Rapid ORF Description & Evaluation Online)[17] and antiSMASH[22] have allowed users to search for RiPP BGCs in a high-throughput manner. Unfortunately, attempts to isolate the RiPPs encoded by these BGCs from their native producers often fail because many of the clusters are silent under standard laboratory conditions, a common problem for natural product discovery[23–25]. One solution to the silent BGC problem is pathway refactoring and heterologous expression in genetically tractable hosts[26]. Indeed, many RiPPs have been obtained via heterologous expression; however, the synthetic biology workflow is experimentally demanding and low throughput[2]. In contrast, biological foundries (biofoundries) such as the Illinois Biological Foundry for Advanced Biomanufacturing (iBioFAB)[27] use robotic automation to perform synthetic biology workflows in high-throughput[27]. Although biofoundries have not yet been applied to access the products of silent BGCs in the natural product field, they have already shown great power in the automation of DNA assembly[28], which is a pivotal step for BGC cloning/refactoring and heterologous expression.

In this study, we develop a <u>F</u>ast, <u>A</u>utomated, <u>S</u>calable, high-<u>T</u>hroughput pipeline for RiPP discovery (FAST-RiPPs) by integrating the genome-mining tool RODEO for identifying RiPP BGCs of interest, iBioFAB for automated pathway refactoring, and heterologous expression in *E. coli* for RiPP production (Fig. 1c). We demonstrate the capability of FAST-RiPPs by surveying 96 RiPP BGCs from eleven different RiPP classes, leading to the isolation of 30 molecules representing six RiPP classes, including three that possess antimicrobial activity against certain ESKAPE pathogens.

## Results

### Overview of FAST-RiPPs

To access RiPPs possessing scaffolds with the potential for antimicrobial activity, we first used the high-throughput genome-mining tool RODEO[17] to identify and prioritize 96 putative RiPP BGCs that are from various bacterial phyla and possess precursor peptide sequences distinct from previously characterized RiPPs (Supplementary Data 1). Detailed criteria used for BGC selection are provided in the Supplementary Information. Upon prioritization of BGCs, synthetic genes codon-optimized for *E. coli* were obtained and refactored via a high-throughput method on the iBioFAB[28]. Taking advantage of the direct gene-encoded nature of RiPP precursor peptides, we introduced an N-terminal His$_6$-tag to each so that the modified peptide could be purified by immobilized metal affinity chromatography (IMAC). For cases where multiple precursor peptides are encoded, one of two approaches was taken: (i) the precursor peptides were refactored into the same plasmid with individual His$_6$-tags, or (ii) the precursor peptides with individual His$_6$-tags were refactored into separate plasmids

with each encoding the complete set of biosynthetic proteins. Genes that function in transcriptional regulation, leader peptide removal, and cellular export were omitted from refactoring. Therefore, the RiPP products obtained in this work would likely be retained inside the producing cell, contain the His[6] purification handle, and reduce the chance of host toxicity (most RiPPs are inactive until leader peptide removal[29]). Lasso peptides and thiopeptides include leader peptide removal as an obligatory biosynthetic step[2], thus they represent exceptions to our otherwise common workflow. All successfully refactored RiPP BGCs were expressed in *E. coli* and matrix-assisted laser desorption/ionization time-of-flight mass spectrometry (MALDI-TOF MS) was used to detect the desired modified peptides (Fig. 1).

## High-throughput refactoring of RiPP BGCs

We developed an automated high-throughput workflow for pathway refactoring based on Golden Gate assembly and iBioFAB[28]. The Golden Gate method joins multiple DNA fragments linked by orthogonal 4-bp linkers via Type IIS restriction enzymes and T4 DNA ligase[30]. Adapting a previous design[31–33], nine helper plasmids and one receiver plasmid were constructed to refactor up to nine genes in a single *Bsa*I-catalyzed reaction (Fig. 2). Each helper plasmid contained two *Bsa*I recognition sites flanking the insert, followed by a unique pair of 4-bp linkers, a T7 promoter, and a T7 terminator, between which codon-optimized genes encoding the precursor peptide and biosynthetic proteins lacking *Bsa*I sites were separately inserted. Given the variable number of genes in RiPP BGCs, two versions of the helper plasmids were created that contained a 4-bp linker either complementary to the next helper plasmid in the assembly order or complementary to the backbone linker. After one-pot Golden Gate assembly, successful refactoring was assessed by blue-white screening and confirmed by Sanger DNA sequencing.

The iBioFAB[28] is an integrated robotic system that enables rapid prototyping of biological systems (Supplementary Fig. 1). To manage automation processes, *Momentum* software was used to communicate with devices, control the robotic arm, and send commands to devices according to user input. Design of the refactoring workflow started with well-defined unit operations such as liquid handling, centrifugation, and temperature control that were operated by the corresponding instruments on the iBioFAB (Supplementary Fig. 1). Sample location and transportation routes between unit operations were defined and matched with component instruments in the *Momentum* software. Pipetting tasks were performed using a Labcyte Echo 550 (acoustic nanoliter-scale dispensing system) for setting up 96 Golden Gate reactions and a Tecan FluentControl 1080 (multipurpose liquid handling system) for *E. coli* transformation and plasmid extraction (Fig. 2). Constructs were verified by high-throughput gel electrophoresis analysis of restriction enzyme digests of isolated plasmids using an Agilent Fragment Analyzer (see Methods).

Using the high-throughput RiPP BGC refactoring workflow, 83 out of 96 chosen RiPP BGCs were successfully refactored (Supplementary Fig. 2). The target BGCs contained two to nine genes with a maximum length of 18 kb. BGCs containing fewer than five biosynthetic genes were refactored with a 100% success rate while BGCs with more genes had lower success rates. For example, the thiopeptide BGCs contained between six and nine genes and had a success rate of 12.5% (Fig. 3). The assembly fidelity decreased when refactoring more than three biosynthetic genes. All incorrect constructs were determined to be partially assembled with various genes omitted. The factors contributing to gene omission during assembly may include: (i) ligation of mismatched pairs of 4-bp linkers, which correlated positively with the number of linkers needed; and (ii) increased errors with more copies of the T7 promoter and terminator sequences in larger BGCs. Accordingly, we observed on three separate occasions (code-named: LanIII-6, LanI-101, and Epi-1; see FAST-RiPPs identifiers in Supplementary Data 1) that BGCs containing two precursor peptides only harbored one

precursor gene in the refactored plasmid, despite all the other genes being correctly assembled. In the case of LanIII-6, synthetic DNA for the second precursor was co-expressed in a separate plasmid with the refactored plasmid. For LanI-101 and Epi-1, the refactored plasmids were evaluated for expression with just the single precursor peptide.

## Production of target RiPPs

Refactored BGCs were heterologously expressed in *E. coli* BL21(DE3) using four cultivation conditions with varied temperatures and media (see Methods). The peptides produced were purified by IMAC and analyzed by MALDI-TOF MS (Fig. 4). Exceptions were the lasso peptides and thiopeptides, as the leader peptide, and therefore the His[6]-tag, is removed during biosynthetic maturation. For these refactored BGCs, peptides were isolated using methanol extraction of the cell pellets followed by MALDI-TOF MS analysis. For 27 out of the 83 successfully refactored BGCs, masses corresponding to modified precursor peptides were observed. These peptides were from the lanthipeptide (PTM: (methyl)lanthionine, labionin, dehydroamino acids), glycocin (PTM: *S*, *O*-glycosylation of Cys/Ser), linear azol(in)e-containing peptide (LAP) (PTM: Cys, Ser, or Thr derived azol(in)es), graspetide (PTM: macrolactones/lactams), lasso peptide (PTM: macrolactam with threaded C-terminal tail), and thioamitide (PTM: backbone thioamide) classes of RiPPs. For BGCs that contained a precursor peptide with a double Gly recognition motif common to substrates for peptidase-containing ATP-binding transporters (PCATs), such as nitrile hydratase leader peptide (NHLP)/Nif11 family precursor peptides, the substrate tolerant C39 protease LahT150 was used to obtain the modified core peptides[34,35]. For BGCs that encoded proteases other than PCATs, synthetic genes for these proteases were obtained, expressed, and the enzymes used in cleavage assays to obtain the modified core peptides. When a protease was missing in the BGC, but the leader removal site could be predicted for the substrate peptide, commercial proteases were used to obtain smaller peptide fragments. Further analyses were then carried out to locate the site(s) of modification. Finally, for cases where the protease cut site could not be predicted, commercial proteases were used to obtain fragments of appropriate sizes for high-resolution electrospray ionization tandem mass spectrometry (HR-ESI MS/MS) analysis, but the modified peptides were not carried forward for bioactivity screening and full structural characterization. Modified core peptides were tested for growth suppression activities against a panel of ESKAPE pathogens. Those that displayed antimicrobial activities were selected for detailed structural characterization (Fig. 4).

## Production of lanthipeptides with unique ring connectivities

Lanthipeptides are attractive targets for discovering bioactive compounds to combat infections caused by clinically relevant Gram-positive bacteria[36]. The promise of lanthipeptides as antimicrobials is partly due to their unique modes of action and their high potency[37–41]. For example, the class I lanthipeptide nisin (Fig. 1) has been used in the food industry for over 50 years with little development of resistance[38]. The class-defining features of lanthipeptides are (methyl)lanthionine linkages installed by lanthipeptide synthetases in a two-step mechanism[42]. First, Ser and Thr residues are dehydrated, followed by a cyclization reaction where the thiol groups of Cys residues are added to the dehydrated residues. Lanthipeptides are subdivided into five classes (I-V) based on the enzymatic machinery used to install the lanthionine structures[2,43]. When this study was initiated, only four classes of lanthipeptides were known (I-IV), with a recent bioinformatic study identifying 8500 unique lanthipeptide precursor peptides grouped based on their core peptide sequence[18]; many lanthipeptide groups do not have any characterized members, and such examples were prioritized in the current study. We used the classification scheme established in the previous bioinformatic study where a

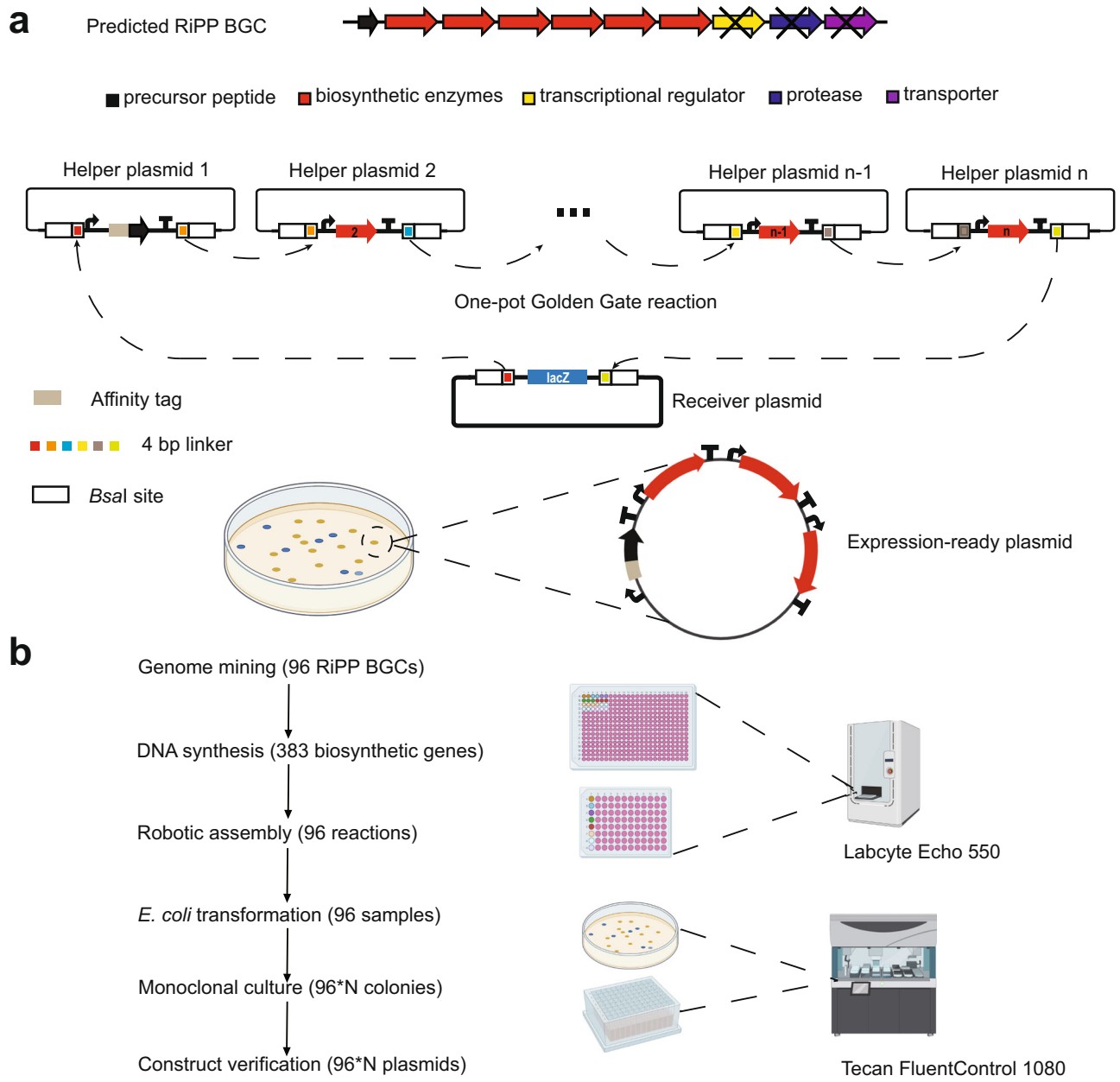

**Fig. 2 | Design of high-throughput pathway refactoring. a** Refactoring scheme for a representative RiPP BGC using Golden Gate assembly. Genes involved in transcriptional regulation, leader peptide removal, and cellular export were omitted from refactoring. The codon-optimized gene encoding the precursor peptide was inserted into helper plasmid 1 with an N-terminal His$_6$-tag and other genes encoding biosynthetic enzymes were inserted into the following helper plasmids. Each helper plasmid contained two *Bsa*I recognition sites flanking the insert, followed by a unique pair of 4-bp linkers, a T7 promoter, and a T7 terminator. The constructed cassettes that comprise of a gene driven by a T7 promoter in the helper plasmid can then be assembled in a defined order into the *lacZ*-containing receiver plasmid with pET28a backbone in a single step via *Bsa*I-catalyzed Golden Gate assembly. The transformants were assessed by blue-white screening and further confirmed by Sanger DNA sequencing. **b** Workflow for high-throughput refactoring of 96 RiPP BGCs. Labcyte Echo 550 was used for setting up 96 Golden Gate reactions and Tecan FluentControl 1080 was used for *E. coli* transformation and plasmid extraction. N represents the number of constructs evaluated for each refactored RiPP BGC. Figure 2b was created with BioRender.

number after the class denotes the group number. The absence of a number indicates that the BGC was not part of this previous study.

To access lanthipeptides with unique ring connectivities, we selected seven class I (LanI), thirteen class II (LanII), seven class III (LanIII), and seven class IV (LanIV) lanthipeptide BGCs. Upon screening each refactored BGC without any optimization efforts, one LanI (14%), nine LanII (64%), three LanIII (42%) and two LanIV (29%) BGCs produced modified peptides (Table 1), whereas the other BGCs did not produce modified peptides detectable by MALDI-TOFMS. The degree of cyclization of each modified lanthipeptide was examined using

*N*-ethylmaleimide (NEM) alkylation and dithiothreitol (DTT) assays that probe for free Cys residues and dehydroamino acids, respectively[44]. Based on these assays, the LanI-101 refactored BGC from *Runella limosa* produced a mixture of products with four to six dehydrations (Table 1). Glutathione adducts were also observed by MALDI-TOF MS (Supplementary Fig. 3). Owing to the irreversible nature of glutathione addition to dehydroamino acids at physiological pH, the desired product could not be isolated for bioactivity screening and structural characterization. Fortunately, glutathione addition was not a commonly encountered problem for lanthipeptide production.

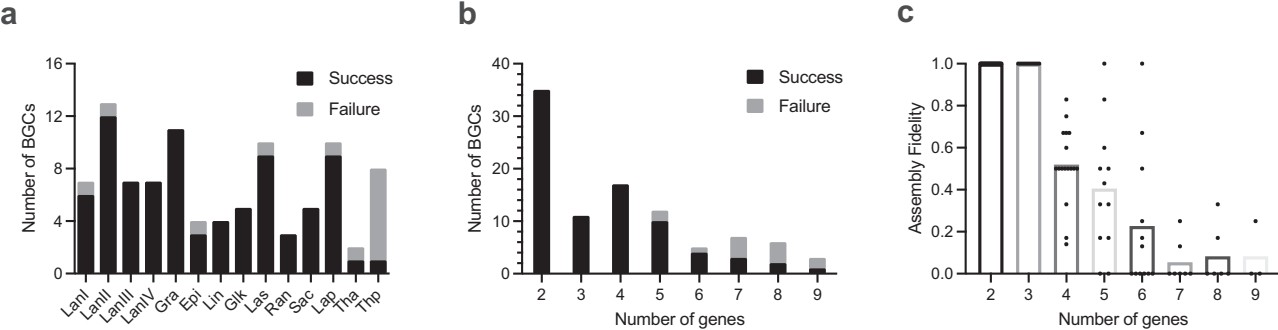

**Fig. 3 | Success rate and fidelity of pathway refactoring. a** Results of refactoring among different investigated RiPP classes. LanI, class I lanthipeptide; LanII, class II lanthipeptide; LanIII, class III lanthipeptide; LanIV, class IV lanthipeptide; Gra, graspetide; Epi, epipeptide; Lin, linaridin; Glk, glycocin; Las, lasso peptide; Ran, ranthipeptide; Sac, sactipeptide; Lap, linear azol(in)e-containing peptide; Tha, thioamitide; Thp, thiopeptide. Effect of the number of genes on **b** refactoring success rate and **c** assembly fidelity. Data are presented as the mean for assembly fidelity ($n_2 = 35$, $n_3 = 11$, $n_4 = 17$, $n_5 = 12$, $n_6 = 5$, $n_7 = 7$, $n_9 = 6$, $n_8 = 3$, where $n_x$ denotes the number of data points for group x) and are calculated as the ratio of the number of correct constructs to the total number of constructs evaluated. The automated refactoring of RiPP BGCs was performed only once.

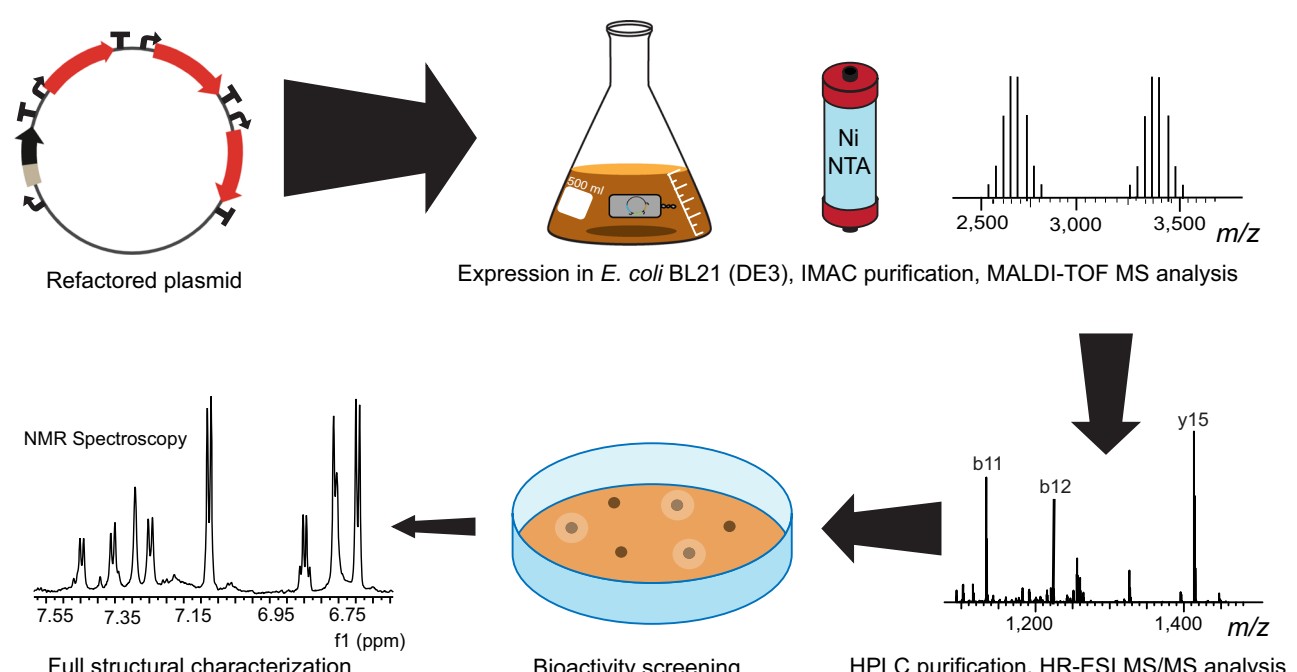

**Fig. 4 | Schematic FAST-RiPPs workflow for rapid discovery of RiPPs.** Refactored plasmids were used to transform *E. coli* BL21 (DE3) for expression. Peptide products were isolated by IMAC purification or methanol extraction and analyzed by MALDI-TOF-MS. Peptides with modifications were purified by reversed-phase HPLC and analyzed by HR-ESI-MS/MS to locate the site(s) of modification. Leader peptides were subsequently removed by protease digestion and the modified core peptides were screened for growth suppression activity against the panel of ESKAPE pathogens. The structures of bioactive compounds were further characterized by NMR spectroscopy.

For the successfully produced and modified LanII peptides, all Cys residues in the ten modified peptides were involved in ring formation except for LanII-2B, with varying numbers of dehydration for each peptide (see Table 1, Fig. 5, Supplementary Figs. 4-19). LanII (Fig. 5) and LanII-23 (Supplementary Fig. 4a) are the first characterized lanthipeptides from gammaproteobacteria, and LanII-2B derived from *Archangium violaceum* is the first characterized lanthipeptide from a deltaproteobacterium (Supplementary Fig. 4b). The peptides from LanII-2A and LanII-2C were dehydrated three and five times, respectively, and fully cyclized (Fig. 5b, Supplementary Fig. 4c, 7 and 13), whereas the precursor peptide from LanII-2B was dehydrated twice, with two Cys that were cyclized and a third present as a free Cys as evidenced by NEM/DTT assays (Table 1, Supplementary Figs. 4b, 11) and HR-ESI MS/MS data (Supplementary Fig. 12).

When LanII-2D from *Streptomyces* sp. ADI96-02 was initially chosen for this study, it was an orphan cluster encoding two precursor peptides with dissimilar core peptides and two LanM synthetases. However, as this work was ongoing, the products of a BGC in *Streptomyces roseosporus* NRLL11379 were reported that have identical core peptides as LanII-2D and were shown to constitute a two-component lantibiotic with synergistic antimicrobial activity against methicillin-resistant *S. aureus* (MRSA) and vancomycin-resistant enterococci (VRE)[45]. Therefore, although the LanII-2D peptides were fully modified in *E. coli* in the current work (Supplementary Fig. 4d), they were not investigated further. Like LanII-2D, the LanII-57 BGC from the Actinomycetota *Kitasatospora xanthocidica* encodes two precursor peptides, A and B, with dissimilar core peptides, but this time with a single class II lanthipeptide synthetase. Both precursor peptides were dehydrated

**Table 1 | Summary of produced modified peptides across Bacteroidota, Pseudomonadota, Cyanobacteriota, Actinomycetota and Bacillota**

| FAST-RiPPs identifier[a] | Phylum | Bacterial Strain | Modifications |
|---|---|---|---|
| LanI-101 | Bacteroidota | *Runella limosa* | 4–6 dehydrations; 3 (methyl)lanthionines |
| LanII[b] | Pseudomonadota | *Dyella* sp. 333MFSha | 4 dehydrations; 3 (methyl)lanthionines |
| LanII-23 | Pseudomonadota | *Pseudomonas ogarae* | 5 dehydrations; 4 (methyl)lanthionines |
| LanII-2A | Cyanobacteriota | *Scytonema hofmannii* | 3 dehydrations; 2 (methyl)lanthionines |
| LanII-2B | Pseudomonadota | *Archangium violaceum* | 2 dehydrations; 2 (methyl)lanthionines |
| LanII-2C | Cyanobacteriota | *Coleofasciculus chthonoplastes* | 5 dehydrations; 5 lanthionines |
| LanII-2D | Actinomycetota | *Streptomyces* sp. ADI96-02 | 9 dehydrations; 6 (methyl)lanthionines |
| LanII-2E | Cyanobacteriota | *Desertifilum* sp. IPPAS B-1220 | 1 dehydration; 1 (methyl)lanthionine |
| LanII-56 | Actinomycetota | *Streptomyces* sp. NRRL S-350 | 3 dehydrations; 3 (methyl)lanthionines |
| LanII-57 | Actinomycetota | *Kitasatospora xanthocidica* | (A) 7 dehydrations; 5 (methyl)lanthionines (B) 4 dehydrations; (methyl) lanthionines |
| LanIII[b] | Pseudomonadota | *Myxococcus fulvus* | 7 dehydrations; 3 (methyl)lanthionines |
| LanIII-6 | Bacillota | *Bacillus amyloliquefaciens* | (A) 6 dehydrations; 1 labionin (B) 7 dehydrations; 1 labionin |
| LanIII-43 | Bacillota | *Bacillus cereus* | 5 dehydrations; 2 labionins |
| LanIV[b] | Pseudomonadota | *Chondromyces crocatus* sp. Cm c5 | 1 dehydration; 1 methyllanthionine |
| LanIV[b] | Actinomycetota | *Streptomyces* sp. DvalAA-43 | 3 dehydrations; 2 (methyl)lanthionine |
| Lap-1 | Bacillota | *Enterococcus caccae* | 8 thioazol(in)es |
| Glk-1 | Bacillota | *Bacillus cereus* | 1 glycosylation |
| Glk-2 | Bacillota | *Enterococcus faecalis* ATCC 6055 | 2 glycosylations |
| Glk-3 | Bacillota | *Dellaglioa algida* | 3 glycosylations |
| Las-2 | Bacillota | *Bacillus cereus* | 1 macrolactam |
| Las-6 | Actinomycetota | *Streptacidiphilus melanogenes* | 1 macrolactam |
| Gra-3 | Bacillota | *Bacillus* sp. S66 | 3 macrolactams/macrolactones |
| Gra-4 | Pseudomonadota | *Burkholderia seminalis* | 2 macrolactams, 2 macrolactones |
| Gra-5 | Pseudomonadota | *Glaciecola* sp. KUL10 | 1 macrolactam/macrolactone |
| Gra-7 | Pseudomonadota | *Legionella pneumophila* | (A) 3 macrolactams/macrolactones (B) 3 macrolactams/macrolactones |
| Gra-8 | Pseudomonadota | *Lysobacter capsici* | 3 macrolactams/macrolactones |
| Tha-1 | Pseudomonadota | *Desulforegula conservatrix* | 1 thioamidation |

[a]For NCBI accession IDs, see Supplementary Dataset. [b]These lanthipeptide BGCs were not annotated in the previous bioinformatic study[18], and therefore do not have a group number.
LanI, class I lanthipeptide; LanII, class II lanthipeptide; LanIII, class III lanthipeptide; LanIV, class IV lanthipeptide; Gra, graspetide; Glk, glycocin; Las, lasso peptide; Lap, linear azol(in)e-containing peptide; Tha, thioamitide.

and cyclized by this LanM in *E. coli* (Table 1, Supplementary Figs. 4e, 15–17). LanII-2E resulted in a product consistent with a single dehydration (Supplementary Fig. 4f), even though the core sequence of the precursor peptide contains seven Ser/Thr residues and four Cys residues, suggesting that it is only partially modified.

The refactored class III lanthipeptide BGCs from *Bacillus cereus* (LanIII-43), *Myxococcus fulvus* (LanIII), and *Bacillus amyloliquefaciens* (LanIII-6) all resulted in modified peptides (Table 1, Supplementary Figs. 20–25). The HR-ESI MS/MS fragmentation pattern for the five-fold dehydrated LanIII-43 suggests the presence of two labionins[46] in the modified peptide (Supplementary Fig. 23). The LanIII BGC does not encode a protease and the peptide substrate lacked a clearly defined site for leader peptide removal, and thus the seven-fold dehydrated product was not pursued further. As this work was ongoing, the structures of the LanIII-6 peptides were reported by another research group showing very similar data (Supplementary Fig. 22)[47], and therefore they were also not characterized further.

Two LanIV BGCs from the deltaproteobacterium *Chondromyces crocatus* sp. Cm c5 and Actinomycetota *Streptomyces* sp. DvalAA-43 also resulted in the production of modified peptides (Table 1). The refactored *C. crocatus* BGC produced a peptide that was dehydrated and cyclized only once whereas the product from the *Streptomyces* BGC was dehydrated three times and cyclized twice as evidenced by DTT and NEM assays and MALDI-TOF MS (Supplementary Figs. 26, 27, 29). HR-ESI MS/MS analysis confirmed the presence of a single

methyllanthionine ring localized to the C-terminus for the *C. crocatus* product (Supplementary Fig. 28), whereas the product from *Streptomyces* contained one lanthionine and one methyllanthionine that possibly overlap (Supplementary Fig. 30).

## Production of lasso peptides with unique core peptide sequences

Lasso peptides are another class of RiPPs whose members include antimicrobial compounds[2]. Lasso peptides are characterized by the formation of a macrolactam between the N-terminus and the side chain of an acidic residue in the core peptide. The tail of the peptide is lodged through the plane of the ring, resulting in the class-defining threaded conformation (Fig. 1). Nine refactored lasso peptide BGCs were expressed in *E. coli* under the four standard conditions. Methanolic extracts of the cell pellets were analyzed by MALDI-TOF MS, and two BGCs from *B. cereus* (Las-2) and *Streptacidiphilus melanogenes* (Las-6), produced modified peptides with masses consistent with a single dehydration (Table 1, Fig. 5, Supplementary Fig. 31). HR-ESI MS/MS analysis of the HPLC-purified Las-2 peptide showed fragmentation between residues in the predicted tail region, but not in the predicted ring region. This observation is consistent with the formation of a macrolactam between the N-terminus and Asp8 (Supplementary Fig. 32). For Las-6, HR-ESI MS data were consistent with the predicted mass of the mature lasso peptide (Supplementary Fig. 33); however, no fragmentation was observed in the HR-ESI MS/MS spectrum, possibly

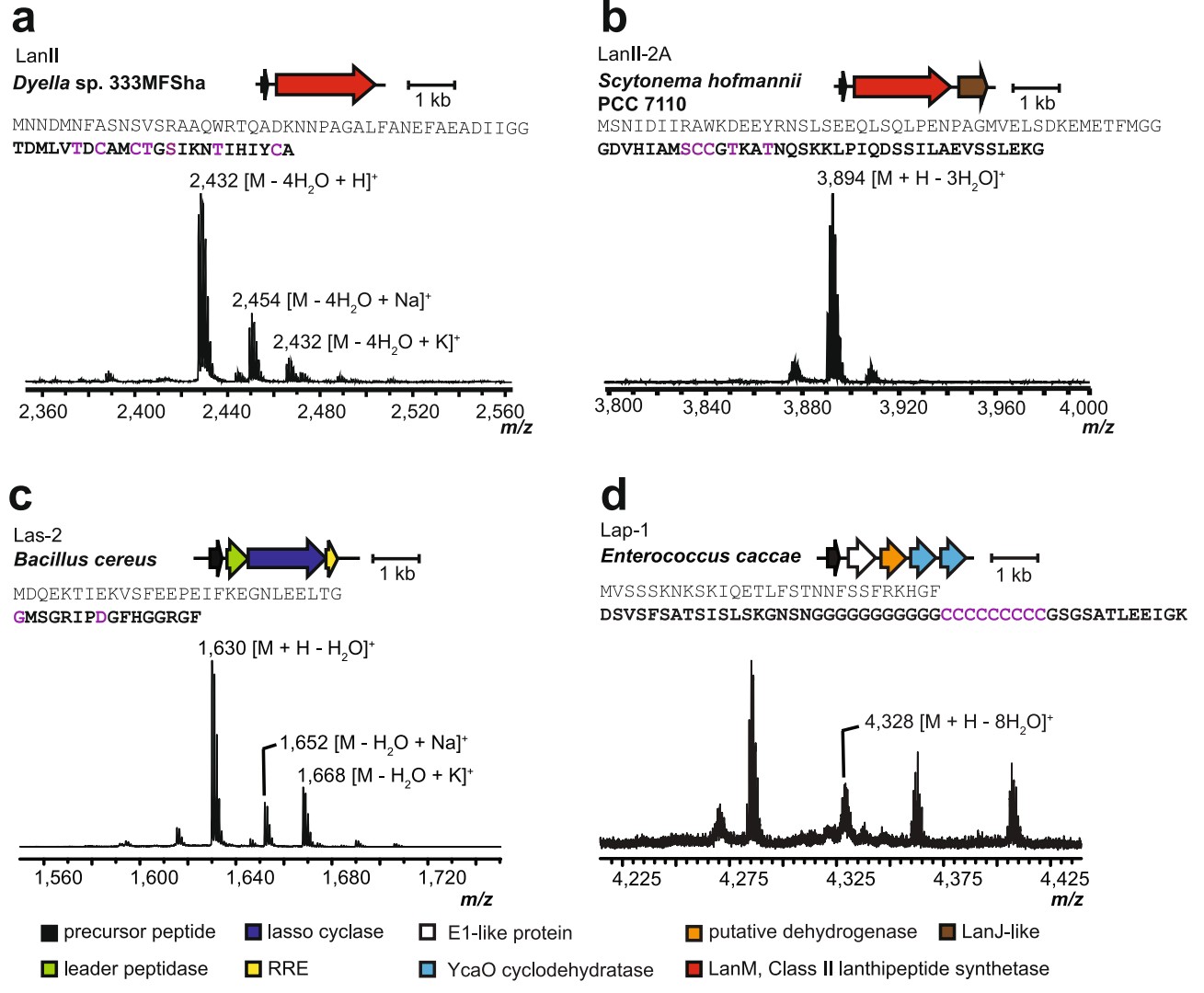

**Fig. 5 | MALDI-TOF MS data of representative products from refactored RiPP BGCs.** Products of the refactored BGCs of **a** LanⅡ, **b** LanⅡ-2A, **c** Las-2, and **d** Lap-1. Shown are the producing organism, gene diagram for the BGC, sequence of the precursor peptide with the predicted core peptide bolded and the location of PTMs in purple, and MALDI-TOF mass spectrum of the modified peptide. Peptides were analyzed after protease digestion with endoproteinase AspN (**a**, **d**) LahT150 (**b**) or after in vivo production and extraction (**c**). RRE, RiPP recognition element.

due to low abundance. Yield optimization is in progress to facilitate further characterization of these peptides.

**Production of graspetides with unique ring connectivities**

Graspetides are defined by the ATP-grasp ligase-dependent formation of macrolactone and macrolactam linkages between the side chains of donor Ser, Thr, or Lys residues and acceptor Asp or Glu residues[2]. Recent bioinformatic studies identified ~4000 graspetide BGCs, split into 24 groups based on sequence similarity of the ATP-grasp enzymes and the presence of conserved motifs within the predicted precursor peptides[14,48,49]. Eleven graspetide BGCs were refactored in this study and expressed under the four standard conditions. For five of the BGCs, MALDI-TOF MS analyses demonstrated the production of modified peptides with varying levels of dehydration (Supplementary Fig. 34, Table 1). We next sought to determine residues modified in each of these peptides. Because the modified residues are often conserved amongst related graspetides[48], attempts were made to bioinformatically predict the modified residues. For each graspetide, the precursor was aligned with the precursor peptides from the ten most related graspetide BGCs. For Gra-5, the alignment revealed conservation of two acceptor residues and at least two donor residues

(Supplementary Fig. 35), but only one dehydration was observed in the expression system (Supplementary Fig. 34d). For Gra-8, the four most C-terminal acceptor residues were conserved across the most related peptides, and each peptide contained at least four C-terminal donor residues (Supplementary Fig. 36), but a maximum of three dehydrations was observed (Supplementary Fig. 34e). Based on this bioinformatic analysis, these two peptides appear inefficiently modified, and therefore they were not characterized further. In the case of Gra-3, the most closely related precursor peptides contained four acceptor residues and a series of four contiguous donor residues, and an additional conserved Lys residue at the C-terminus (Supplementary Fig. 37). The peptide was digested with trypsin and a mass consistent with a three-fold dehydrated C-terminal fragment was observed in the MS spectra. It is possible that this peptide is only partially modified, but it is similar in sequence to marinomonasin[50], which is also dehydrated three times. The HR-ESI MS/MS spectrum of the trypsin digested C-terminal fragment showed minimal fragmentation, precluding further localization of the rings (Supplementary Fig. 38).

The Gra-4 product was dehydrated four times and contains two TxxKxxDD repeats, and the most closely related precursor peptides also contain at least one such repeat (Supplementary Fig. 39).

Graspetides often contain repeats in their core peptide (e.g., pleisiocin[51]), and thus the TxxKxxDD repeats are likely the sites of modification. To test this hypothesis, the precursor peptide was digested with AspN, and reacted with sulfosuccinimidyl-acetate to probe for free amines. The C-terminal fragment was singly labeled, suggesting that the Lys side chains of the peptide were involved in macrolactam bonds, and the only free amine was the N-terminus, consistent with the TxxKxxDD repeats being modified (Supplementary Fig. 40). The HR-ESI MS/MS spectrum of a trypsin fragment of the peptide was also consistent with the predicted sites of modification, as no fragments were observed in either KxxDD region (Supplementary Fig. 41). Peptidic macrolactones can undergo a McLafferty rearrangement in the mass analyzer, resulting in modified Thr residues being observed as dehydrated[48]. Consistent with this possibility, fragmentation was observed between each conserved Thr and Lys, with dehydrations localized to each Thr residue. These data are insufficient to determine which Asp forms the macrolactam and which forms the macrolactone. However, fragmentation was observed in the region separating the two repeats, suggesting that the bonds are formed within each repeat, not between the repeats. Gra-7 contains two precursors, which were refactored into separate plasmids together with the same ATP-grasp enzyme. After expression, the isolated peptides were dehydrated three times, and subsequent digestion with LahT150 localized these modifications to the predicted core region of precursor A, however the digestion with precursor B was unsuccessful. Low yields from the heterologous expression prevented HR-ESI MS/MS analysis of either peptide. Future work will focus on improving these yields to allow for further structural characterization.

## Production of thioamitides

Thioamitides are RiPPs containing thioamides formed by YcaO and TfuA proteins, sometimes in concert with ThiF and ThiS homologs[2]. Two thioamide clusters were selected for refactoring, both containing multiple additional enzymes likely producing secondary modifications. Only Tha-1 was successfully refactored, and upon expression it was observed with a mass consistent with the formation of a single thioamide as evidenced by MALDI-TOF MS (Supplementary Fig. 42). HR-ESI MS/MS confirmed the presence of a thioamide in Tha-1, but its position within the peptide could not be located (Supplementary Fig. 43). The Tha-1 BGC also encodes a glycosyltransferase and a methyltransferase that are not required for the formation of thioamides but may further modify the precursor peptide. Since no additional modifications were observed, the isolated peptide was not carried forward for bioactivity screening and detailed structural characterization.

## Production of linear azol(in)e-containing peptides

Linear azol(in)e-containing peptides (LAPs) constitute another major class of RiPPs, which are characterized by the presence of azole or azoline heterocycles formed through cyclodehydrations and dehydrogenations of Cys/Thr/Ser within a linear peptide[52]. To date, 17 LAPs are known, most of which show cytolytic activity (streptolysin S, clostridiolysin S, listeriolysin S) or narrow-spectrum antibacterial activities by inhibiting DNA-gyrase (microcin B17) or the ribosome (klebsazolicin, phazolicin)[3]. Using the FAST-RiPPs pipeline, nine out of ten BGCs from four different phyla were successfully refactored. Upon screening of the four expression conditions and subsequent MALDI-TOF MS analyses, one BGC, Lap-1 encoding a precursor peptide containing nine Cys residues, produced a modified peptide that was 8-fold dehydrated (Fig. 5d). Labelling of the modified peptide with NEM confirmed the presence of one free Cys residue (Supplementary Fig. 44). Subsequent HR-ESI MS/MS analysis of the AspN-digested peptide produced fragments consistent with the formation of eight thiazol(in)e rings (Supplementary Fig. 45). The modified peptide could not be purified for bioactivity screening and subsequent structural

characterization due to the extremely hydrophobic nature of the core peptide. Based on the Cys-rich core peptide sequence of Lap-1, the function is proposed to be analogous to that of the streptolysin S cytolytic toxin[53].

## Production of glycocins

Glycocins are glycosylated bacteriocins that have characteristic sugar moieties installed on Ser/Thr/Cys by glycosyltransferases[54]. To date, ten different glycocins have been experimentally identified in Bacillota that exhibit narrow-spectrum antibacterial activities against pathogenic bacteria, including *Streptococcus pyogenes*, MRSA and food-spoilage bacteria like *Listeria monocytogenes*[16]. Five BGCs that encode putative glycocins were successfully refactored for heterologous expression in *E. coli*. Three of these BGCs from *B. cereus*, *Enterococcus faecalis* and *Dellaglioa algida*, produced modified peptides Glk-1, Glk-2, and Glk-3, respectively, as determined by MALDI-TOF MS analysis (Supplementary Fig. 46). HR-ESI MS/MS analysis of LahT150-liberated core peptide suggested that Glk-1 is *S*-glycosylated with one N-acetylhexosamine (HexNAc) (Supplementary Fig. 47). However, attempts to obtain the core peptides of Glk-2 and Glk-3 using LahT150 proved futile. HR-ESI MS data of the AspN-digested peptides showed a mass increase of 324 Da for Glk-2 (consistent with modification with two hexoses) and a mass increase of 568 Da for Glk-3 (consistent with the addition of one hexose and two HexNAc moieties) (Supplementary Figs. 48, 49). However, no fragmentation was observed in the HR-ESI MS/MS spectrum of Glk-3. Future studies will focus on expressing and purifying the C39 protease domain encoded in the BGC of Glk-2 and Glk-3 to access the pure core peptides for bioactivity screening and structural characterization.

## Bacterial growth suppression assays and structure determination

Modified peptides which contained a clearly defined site for leader sequence removal were digested with an appropriate protease, purified by HPLC, and lyophilized. The modified core peptides were redissolved in 5% dimethyl sulfoxide, 30% acetonitrile, 10% methanol or water depending on the solubility. The concentrations were estimated and the peptides were evaluated for growth suppression activity against the panel of ESKAPE pathogens using agar diffusion assays. For peptides that produced zones of inhibition, the minimal inhibitory concentrations (MICs) were determined using a serial dilution assay.

The modified core peptides for LanII-2A and LanII were accessed by digesting with LahT150[35] and endoproteinase AspN, respectively (despite containing a possible double Gly-motif, LahT150 did not cleave the LanII peptide). The peptides were both selectively active against *K. pneumoniae* with MICs of 4.1 μM for LanII-2A and 126 μM for LanII (Supplementary Fig. 50). To the best of our knowledge, these represent the first lanthipeptides with anti-*Klebsiella* activity. In contrast to LahT150 digestion of LanII-2A, AspN-digestion of the LanII product likely does not produce the native lanthipeptide and the reported MIC here may be higher than that of the native product.

The core peptide for LanII-2B was accessed by digestion with LahT150. As described above, two of four Ser/Thr were dehydrated, and only two out of three Cys were cyclized. A presumptive incomplete cyclization could reflect inefficient modification; however, the bioactivity screen showed that the core peptide displayed weak activity against *A. baumanii* and *P. aeruginosa* with an MIC of 130 μM (Supplementary Fig. 50). Future studies will focus on reconstituting LanII-2B biosynthesis in vitro to investigate whether further dehydrations/cyclizations can be achieved, which may increase antibacterial activity.

LanII-56 was digested with endoproteinase GluC, and LanII-2C was cleaved with LahT150. Initial bioactivity screening of both core peptides did not show any antimicrobial activity against the panel of ESKAPE pathogens. Attempts to isolate the two modified core peptides

produced from the LanII-57 BGC proved challenging due to the inability of the protease LahT150 to liberate the core peptides. Digestion with the commercial protease GluC resulted in products with two extra Gly residues at the N-terminus of the core peptide of precursor A and missing two Glu residues from the core peptide of precursor B (Supplementary Fig. 4e). When these two products were tested individually or together against the panel of ESKAPE pathogens, no activity was observed, possibly because of poor solubility or inability to access the correct core peptides in the leader peptide removal step. Future experiments will aim to express and purify the C39 protease domain encoded in the BGC to access the core peptides of the correct size for bioactivity and structural characterization. As noted above, the two correctly cleaved products of LanII-2D displayed activity against MRSA and VRE[45].

A trypsin-like protease is encoded in the BGC of LanIII-43 (Supplementary Fig. 20a). Purification and subsequent enzymatic assays showed that this protease cleaved after Lys24 in the peptide of LanIII-43. Although the presumed native core peptide was obtained, no antimicrobial activity was observed. The BGC for LanIII-6 encodes an S8 protease. When this protease was expressed in *E. coli* and purified, the enzyme efficiently cleaved both peptides at a unique site (Supplementary Fig. 21). Bioactivity screening against the panel of ESKAPE pathogens demonstrated growth suppression activity against *E. faecium* and *S. aureus*. These data agree with the findings in a recent study on the same BGC reported by another research group[47]. The products were therefore not further characterized.

To determine the stereochemistry of the Lan and MeLan residues in the bioactive peptides, the modified LanII-2A, LanII-2B, and LanII peptides were hydrolyzed and the amino acids derivatized as described previously[55,56]. Gas chromatography-mass spectrometry (GC-MS) analysis of the derivatized amino acids revealed LL-lanthionine and DL-methyllanthionine in LanII (Supplementary Fig. 51), whereas LanII-2A contained DL-lanthionine and DL-methyllanthionine (Supplementary Fig. 52). GC-MS analysis of the Lan/MeLan in LanII-2B demonstrated the presence of one DL-lanthionine and one DL-methyllanthionine (Supplementary Fig. 53).

To determine the ring connectivity, HPLC purified LanII was analyzed by NMR spectroscopy. Analysis of TOCSY data identified 23 spin systems attributed to the 23-mer core peptide (Supplementary Fig. 54b, Supplementary Table 2). Subsequent NOESY analysis allowed confirmation of the assignments by cross-peak correlation with TOCSY proton signals (Supplementary Fig. 54b). Lanthionine and methyllanthionine cross-links were first identified by NOEs observed between methylene protons across the thioether linkage (-CH₂-S-CH₂-). Thus, two methyllanthionine linkages were identified between residue 6 (formerly Thr6) and Cys11 and between residue 12 (formerly Thr12) and Cys23. The third ring was formed by a lanthionine between Cys8 and Ala14 (formerly Ser14). This assignment was confirmed by HSQC and HMBC data revealing correlations from the β proton of residue 6 to the β carbon of Cys11 for the first ring, the β protons of Cys23 to the β carbon of residue 12 for the second ring and the β protons of Cys8 to the β carbon of Ala14 for the third ring (Fig. 6, Supplementary Fig. 55, and Supplementary Table 3). The NMR data also demonstrated that Thr18 embedded within the larger ring had been converted into Dhb (Supplementary Fig. 54a, 54b and 55d), consistent with the MS/MS and DTT assays. For LanII-2A, NMR spectra displayed broad peaks in the only solvent in which the peptide was soluble (DMSO-*d*6), precluding structural determination (Supplementary Fig. 56).

## Discussion

Over the past few decades, RiPPs have attracted significant attention due to their intriguing biosynthetic machinery and diverse biological functions[2]. Most RiPP BGCs are silent under conventional laboratory conditions, likely owing to tight regulation in their native hosts[57], posing a challenge to accessing their products. One way to obtain the

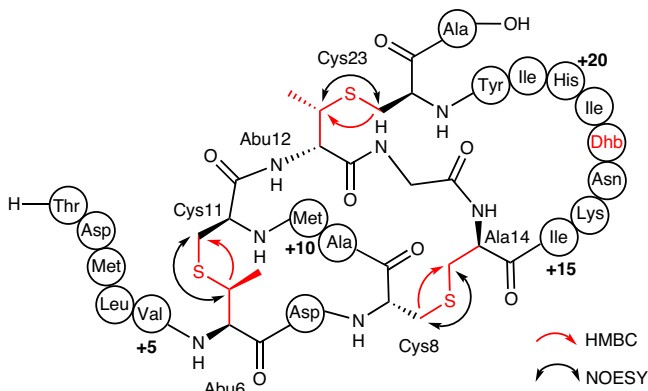

**Fig. 6 | Structure of AspN-digested product of the LanII BGC.** Schematic diagram representing the ring pattern of the LanII product as determined by NMR, GC-MS, and HR-ESI MS/MS analysis. The black double-headed arrows and the red single-headed arrows represent characteristic NOESY and HMBC correlations across the thioether bridges, respectively. The numbers +5, +10, +15 and +20 represent amino acid positions in the core peptide beginning from the N-terminus. Modified amino acid residues are highlighted in red.

products of these silent BGCs is by genetic manipulation to induce gene expression[24,58]. Here, we developed a high-throughput platform named FAST-RiPPs to enable the production of these natural products in a genetically tractable heterologous host. Of the 96 selected RiPP BGCs from different bacterial phyla, 83 were successfully refactored by a biofoundry, as evidenced by Illumina plasmid sequencing. The success rate of 86% highlights the robustness of the refactoring process. The BGCs that failed in the refactoring process were mainly composed of seven or more biosynthetic genes (Fig. 3). The decreased success is partly due to the increasing likelihood of incorrectly ligated linkers in the Golden Gate assembly as the number of biosynthetic genes increases[28].

Screening of four expression conditions for the 83 refactored BGCs resulted in the production of 30 modified peptides from 27 BGCs, while no precursor peptide or no modification could be detected by MALDI-TOF MS for the remaining 56. These 30 modified peptides represent six of the eleven classes that were investigated, with the epipeptide, ranthipeptide, sactipeptide, linaridin, and thiopeptide classes yielding no modified peptides from any of the tested clusters. We note that the unsuccessful classes include those for which a radical SAM (rSAM) enzyme is class-defining (epipeptides[59], ranthipeptides[60], and sactipeptides[61]) or those that have a large number of modification enzymes (thiopeptides[62]) or modifying enzymes that have yet to be reconstituted in *E. coli* (linaridins[63]). rSAM enzymes contain oxidation-sensitive 4Fe-4S clusters, and it is possible that production of modified peptides from rSAM-encoding BGCs failed due to inability to form functional 4Fe-4S clusters. Future work will focus on coexpression of these BGCs with helper plasmids that aid in the formation of stable 4Fe-4S clusters.

Of the 15 lanthipeptide BGCs that produced modified peptides, 60% were from class II. None of the successfully refactored class I BGCs produced efficiently modified peptides. In the biosynthesis of class I lanthipeptides, a LanB dehydratase utilizes glutamyl-tRNA to dehydrate LanA precursor peptides[64,65], and previous studies have shown that some LanBs require tRNA^Glu and glutamyl-tRNA transferase from the native producer for activity[55,66]. The reason for this low success rate is therefore likely the absence of tRNA^Glu and glutamyl-tRNA synthetases from the producing organisms in our current heterologous coexpression system.

Modified peptides were produced from nine class II lanthipeptide BGCs across different phyla including Actinomycetota, Pseudomonadota, and Cyanobacteriota. Of the nine modified peptides, three

yielded core peptides that were bioactive against certain ESKAPE pathogens. Of these three, the modified peptides of LanII-2A and LanII showed antimicrobial activity against *K. pneumoniae*. The precursor peptide of LanII-2A is annotated in NCBI as mersacidin/lichenicidin family type 2 lantibiotic, but an alignment of the precursor peptides for these two known lantibiotics and LanII-2A shows that whereas they indeed have similar leader peptides, their core peptides are considerably different and the lanthipeptide products will not be similar based on the positions of the Ser/Thr and Cys residues (Supplementary Fig. 57). The ring pattern of the product of the LanII BGC is unique amongst known lanthipeptides. Orthologs of the precursor peptide are found in several genera of Rhodanobacteraceae and we propose the name rhodanocins. NCBI lists the precursor peptide from *Dyella* sp. 333MFSha (WP_043693394.1) as a member of a domain-of-unknown function (DUF6229; Pfam 19740) with the family reported in GenBank and MIBiG[67] as being part of the BGC for the non-ribosomal peptide lysobactin from *Lysobacter* sp. ATCC 53042. However, our data suggest these are lanthipeptide precursor peptides and unrelated to lysobactin biosynthesis. The protein (WP_036102482.1) annotated as DUF6229 member in the deposited sequence for the lysobactin BGC (JF412274.1)[68] encodes a lanthipeptide with a considerably different predicted ring pattern than the rhodanocins based on the reported sequence (Supplementary Fig. 58). Thus, we believe that DUF6229 represents at least two different groups of lanthipeptides with different ring patterns. Importantly, these are two more examples of gene annotation and classification of domains-of-unknown function that are mostly based on the leader peptide and that generate incorrect annotations for the RiPPs.

Several other refactored class II lanthipeptide BGCs provided highly modified products (Table 1), but because of the lack of a recognizable leader peptide cleavage site or the absence of a protease in the BGC that could inform on the cleavage site, their bioactivity was not determined. Importantly, the FAST-RiPPs workflow provided the modified peptides that can now be investigated further.

The FAST-RiPPs workflow also allowed for the expansion of class III lanthipeptides across different phyla, including Pseudomonadota and Bacillota. Of the seven class III candidates screened, three refactored BGCs produced modified peptides. The BGC from *B. amyloliquefaciens* produced highly modified peptides that were bioactive against *E. faecium* and *S. aureus*, as has also been reported elsewhere[47]. The production and characterization of two class IV lanthipeptides also expands the diversity of this class from the two previously characterized members, venezuelins[69] and SflA[70], doubling the number of class IV lanthipeptides characterized to date and expanding the phylum origin beyond Actinomycetota. The failure of most class III and IV lanthipeptides to be successfully produced in *E. coli* is likely a result of endogenous proteases as previously reported[70–72].

In the graspetides class of RiPPs, only nine of the 24 bioinformatically predicted groups of graspetides have characterized members[14]. Using the FAST-RiPPs pipeline, graspetides were obtained from groups 8 (Gra-7A and Gra-7B), group 19 (Gra-8), and an unnumbered group (Gra-4), expanding the number of characterized groups to 11. Additionally, one graspetide from group 7, Gra-3, was obtained which contains a series of acceptor and donor residues reminiscent of marinomonasin[50]. Furthermore, one graspetide from group 2, Gra-5, was obtained which lacks one set of putative donor and acceptor residues compared to the prototypical group 2 graspetide, plesiocin.

In summary, we developed an automated high-throughput platform for rapid identification of natural products that are of ribosomal origin across different phyla including Actinomycetota, Cyanobacteriota, Pseudomonadota, Bacillota, and Bacteroidota[73]. The automated process using a biofoundry allowed cloning of 96 BGCs in a single run with a success rate of 86%. Unoptimized screening of four growth conditions yielded 30 products from six RiPP classes, three of which exhibit antimicrobial properties against ESKAPE pathogens and four

more were bioactive peptides recently reported by other research groups. Collectively, the robustness of the approach highlights its potential to rapidly discover lead compounds.

## Methods

### Chemicals, strains, media

Reagents, buffer components, and growth media were purchased from Millipore Sigma (Burlington, MA), Fischer Scientific (Hampton, NH) or Research Products International (Chicago, Illinois) unless otherwise noted. Isopropyl β-D-1-thiogalactopyranoside (IPTG), ampicillin and kanamycin were purchased from Gold Biotechnology (Olivette, Missouri). Oligonucleotides, enzymes, and buffers used for molecular biology were purchased from Integrated DNA Technologies Inc. (Coralville, IA) or New England Biolabs (Ipswich, MA). Synthetic genes were purchased from Twist Biosciences (San Francisco, CA). Plasmid isolation was conducted using QIAprep spin columns according to the manufacturer's protocol (Qiagen, DE). DNA sequencing was performed at the University of Illinois Biotechnology Center (Urbana, IL). *E. coli* NEB-alpha was used for cloning, selection, and plasmid propagation while *E. coli* BL21 (DE3) was used for peptide/protein expression. HiTrap Ni-NTA resin was purchased from Cytiva (Marlborough, MA) and ZipTips containing 0.6 μL of C18 resin were purchased from EMD Millipore (Burlington, MA).

### Pathway refactoring

Pathway construction consisted of 20 μL Golden Gate reactions[74] with 100 ng of the receiver plasmid, and equimolar amounts of each helper plasmid with the corresponding biosynthetic gene, 2 μL of 10× T4 DNA Ligase Reaction Buffer (New England Biolabs, Ipswich, MA), 0.8 μL of BsaI-HF (20,000 U/mL), 0.2 μL of T4 DNA Ligase (2,000,000 U/mL), and ddH$_2$O to a final volume of 20 μL. The reaction mixture was then incubated at 37 °C for 5 min, followed by 30 cycles of 37 °C for 5 min and 16 °C for 10 min, 16 °C for 30 min, 37 °C for 45 min, and 80 °C for 5 min. For chemical transformation, 100 μL of NEB 10-beta competent cells were transformed with 10 μL of the Golden Gate reaction mixture. After recovery in 1 mL of outgrowth medium at 37 °C for 1 h, cells were plated on LB agar plates containing 50 μg/mL kanamycin, 1 mM IPTG and 200 μg/mL X-Gal (for blue/white screening) and incubated overnight at 37 °C. Single white colonies were picked and used to inoculate 5 mL of LB medium supplemented with 50 μg/mL kanamycin for monoclonal plasmid isolation. Correct assembly was first verified by restriction enzyme digestion and finally confirmed by Sanger sequencing.

### High-throughput pathway refactoring using a biofoundry

Helper plasmids containing codon-optimized biosynthetic genes (n = 383) for 96 RiPP BGCs were obtained from Twist Biosciences in a single 384-well polypropylene microplate. Each helper plasmid was diluted with water to 20–40 ng/μL using Multi-Channel Arm (MCA) tips inside FluentControl 1080 liquid handler (Tecan, Männedorf, Switzerland). An online tool (https://cuba.genomefoundry.org/create_assembly_picklists) was used to generate liquid handling picklists containing the pipetting routes. The Echo 550 acoustic liquid handler automatically prepared 96 Golden Gate reactions in a single 96-well plate. The central robotic arm then transferred the plate to the FluentControl 1080 liquid handler to mix the reaction reagents, then the plate was moved to the sealer, centrifuge, and lastly, thermocycler for running the Golden Gate reaction. After that, *E. coli* heat-shock transformation was performed on the FluentControl 1080 liquid handler, which is equipped with a deck capable of holding 25 or more containers, heating/cooling blocks for heat-shock at 42 °C and competent cells incubation at 0 °C, a shaker that allows for cell recovery at 37 °C, and a spiral plater for spreading bacterial cells onto Petri dishes. After overnight incubation, four white colonies from each plate were inoculated into 1.2 mL of LB + kan liquid medium in 2 mL sterile

96-deepwell plates by Flexible-Channel Arm (FCA) tips inside the FluentControl 1080 liquid handler, followed by transferring to the sealer and then to Cytomat_2C2 incubator for overnight incubation at 900 rpm and 37 °C. On the following day, 100 μL of bacterial culture was aspirated and dispensed into a new sterile 96-deepwell plate by MCA tips for to make frozen glycerol stock of correct clones. After spinning down the remainder of the bacterial culture using the Agilent centrifuge, the supernatant was removed by MCA tips. Then high-throughput plasmid extraction using the PureLink® Pro Quick96 Plasmid Kit (Thermo Scientific) was conducted on the FluentControl 1080 following a customized protocol. The vacuum manifold 'TeVacS' on the FluentControl 1080 efficiently removed wash buffer and allowed the final elution of plasmid DNA into a 96-well plate. Following the protocol, consistently high yields (50 μL of ~80 ng/μL plasmids) of high-purity plasmid DNA were be obtained from 1.2 mL of monoclonal *E. coli* cell culture in less than 90 min. All purified plasmids were digested with *Asi*SI and *Mlu*I-HF (NEB) and checked via automated parallel capillary gel electrophoresis using Fragment Analyzer (Agilent Technologies, Santa Clara, CA). Correct plasmids were then collected and further confirmed by Illumina MiSeq sequencing.

## Heterologous expression of modified peptides

*E. coli* BL21 (DE3) electrocompetent cells were transformed with 5 μL of 50 ng/μL of sequence-verified plasmids harboring the precursor peptide(s) and biosynthetic genes and incubated at 37 °C overnight. Single colonies were used to inoculate 5 mL of overnight starter cultures in LB containing 50 μg/mL of kanamycin. These cultures were then used to inoculate four expression cultures under the following conditions:

**Condition 1.** The starter culture was added to 250 mL of TB medium, which was incubated at 37 °C with shaking (220 rpm) until an $OD_{600}$ of approximately 1.0 was reached. Cultures were then cooled on ice for 15 min before addition of IPTG to a final concentration of 0.5 mM. Cultures were placed into an 18 °C incubator and shaken overnight (180 rpm).

**Condition 2.** The starter culture was added to 250 mL of TB medium, which was incubated at 37 °C with shaking (220 rpm) until an $OD_{600}$ of approximately 1.0 was reached. Cultures were then cooled on ice for approximately 15 min before addition of IPTG to a final concentration of 0.5 mM. Cultures were placed into an 18 °C incubator and shaken for 3 h (180 rpm).

**Condition 3.** The starter culture was added to 250 mL of TB medium, which was incubated at 37 °C with shaking (220 rpm) until an $OD_{600}$ of approximately 1.5 was reached, at which point IPTG was added to a final concentration of 0.5 mM. Cultures were placed into a 37 °C incubator and shaken for 3 h (180 rpm).

**Condition 4.** The starter culture was added to 250 mL of LB medium, which was incubated at 37 °C with shaking (220 rpm) until an $OD_{600}$ of approximately 0.6 was reached. Cultures were then cooled on ice for approximately 15 min before addition of IPTG to a final concentration of 0.5 mM. Cultures were placed into an 18 °C incubator and shaken overnight (180 rpm).

For all conditions, cells were harvested by centrifugation at 5500 x *g* for 15 min at 4 °C. The supernatant was discarded, and the cell pellet stored at −80 °C until purification. For expression condition(s) that resulted in the production of modified precursor peptide(s), 2 L cultures were grown under the condition that gave the highest yield of fully modified peptide(s) (Supplementary Table 1).

## Purification of modified peptides

Thawed cell pellet was suspended in 50 mL of lysis buffer (20 mM $NaH_2PO_4$, 500 mM NaCl, 0.5 mM imidazole, 6 M guanidine HCl, pH 7.5) per 10 g of cell paste, and stirred at 4 °C for 30 min. The cells were lysed by either sonication (40% amplitude, 4 s pulse, 9.9 s pause, 15 min) or a high-pressure homogenizer (Avestin, Inc.) and the lysate was subjected to centrifugation at 20,000 x *g* for 1 h at 4 °C. The supernatant was further clarified through a 0.45 μm syringe filter prior to loading onto Ni-nitrilotriacetic acid (NTA) resin equilibrated with lysis buffer. After loading the filtered sample, the resin was washed with two column volumes (CV) each of washing buffer 1 (4 M guanidine hydrochloride, 20 mM $NaH_2PO_4$, 300 mM NaCl, 30 mM imidazole, pH 7.5) and washing buffer 2 (20 mM $NaH_2PO_4$, 300 mM NaCl, 30 mM imidazole, pH 7.5). The desired modified peptide(s) were eluted using 1–3 CV of Elution Buffer (20 mM $NaH_2PO_4$, pH 7.5, 100 mM NaCl, 1 M imidazole). A small sample of the eluted peptide was applied to a C18 ziptip and analyzed by matrix-assisted laser desorption/ionization time-of-flight mass spectrometry (MALDI-TOF-MS) to check for expected modifications. The eluted peptides were then desalted using either HPLC purification or Bond Elut C18 solid phase extraction cartridges (Agilent), which were washed with 0.1% trifluoroacetic acid (TFA) in water, and eluted in 80% acetonitrile (MeCN) in water containing with 0.1% TFA. Desalted His-tagged peptide solutions were subsequently lyophilized.

## *N*-Ethylmaleimide (NEM) cysteine alkylation assay

The degree of cyclization of modified lanthipeptides was determined through the alkylation of unreacted Cys thiols with NEM[44]. Prior to alkylation, samples were treated with 10 mM tris(2-carboxyethyl) phosphine (TCEP) in reaction buffer (100 mM sodium citrate, 1 mM EDTA, pH 6) for 30 min to ensure complete reduction of free Cys. NEM was then added to a final concentration of 10 mM. The reaction was allowed to proceed for at least 10 min before desalting using C18-ziptips and analysis by MALDI-TOF MS.

## Dehydroamino acid labeling of modified peptides with dithiothreitol

To determine if the dehydrated amino acids in the modified peptides are involved in lanthionine ring formation, about 200 μM of the modified peptide was added to 100 mM TRIS-HCl buffer pH 7.5, containing 500 mM DTT, and 400 mM diisopropylethylamine (DIPEA). The reaction was allowed to proceed at room temperature for 3 h before desalting using C18-ziptips and analysis by MALDI-TOF MS.

## Amine-specific labeling of modified peptides with sulfo-NHS acetate

To determine the number of free amines in graspetide peptides, about 100 μM of dried, protease-digested peptide was resuspended in 100 μL of 100 mM $NaHCO_3$ pH 8.5 and sulfo-NHS-acetate was added to a final concentration of 5 mM. The reaction was allowed to proceed for 1 h, then the reaction was desalted by ziptip and analyzed by MALDI-TOF MS.

## HPLC purification

After Ni-NTA purification and desalting, modified peptides were purified on an Agilent 1260 II Infinity HPLC, prior to MS/MS analysis, leader peptide removal or bioactivity assays. Conditions used for different compounds are listed below.

**Modified LanAs.** The lyophilized peptides were resuspended in 5–10% Solvent B (0.1% TFA in MeCN) and purified by RP-HPLC using either Phenomenex C5 or C18 column depending on the hydrophobicity (5 μm, 100 Å, 250 mm × 10 mm) with a linear gradient from 2% Solvent B in Solvent A (0.1% TFA in water) to 100% Solvent B over 45 min, monitoring absorbance at 220 nm. Fractions containing His-tagged peptide were identified by MALDI-TOF MS and lyophilized. Formic acid (FA) was used in place of TFA for purification of core peptides for growth-suppression assays.

**Las-2.** The peptide was purified on a reverse phase C18 column (Macherey-Nagel, 250 × 4.6 mm, 5 μm particle size, 300 Å pore size) with a mobile phase of 0.1% FA (A) and MeCN with 0.1% FA. (B) A gradient of 15% MeCN (isocratic, 6 min), 15-33% MeCN (gradient, 25 min), 33–95% MeCN (gradient, 5 min), and 95% MeCN (isocratic, 5 min) at 1 mL/min was used.

**Las-8.** After methanol (MeOH) extraction, the peptide was filtered through a 0.22 μm syringe filter and purified on a reverse phase C18 column (Macherey-Nagel, 250 × 4.6 mm, 5 μm particle size, 300 Å pore size) with a mobile phase of 0.1% aqueous FA and MeCN with 0.1% FA. A gradient of 20% MeCN (isocratic, 6 min), 20-30% MeCN (gradient, 5 min), 30–50% MeCN (gradient, 30 min), 50-90% MeCN (gradient, 2 min), and 95% MeCN (isocratic, 4 min) at 1 mL/min was used.

**Gra-4.** After IMAC and digestion with trypsin, the peptide was desalted using a C18 solid phase extraction column, then purified on a reverse phase C18 column (Macherey-Nagel, 250 × 4.6 mm, 5 μm particle size, 300 Å pore size) with a mobile phase of 0.1% aqueous FA and MeCN with 0.1% FA. A gradient of 20% MeCN (isocratic, 6 min), 20–24% MeCN (gradient, 1 min), 24-32% MeCN (gradient, 44 min), 32–90% MeCN (gradient, 5 min), and 95% MeCN (isocratic, 5 min) at 1 L/min was used.

### Removal of leader peptides
Peptides were digested with either trypsin (Promega), AspN (NEB), LahT150, or GluC (NEB). The LahT150 protease was purified according to previously reported procedures[35]. GluC and AspN reactions were conducted according to the manufacturer's protocol at a 1:100 (w/w) ratio of protease to substrate peptide at 37 °C overnight. Reactions containing LahT150 were carried out in 50 mM Tris HCl (pH 7.5) at a 1:20 (w/w) ratio of protease to substrate at room temperature overnight. Reactions were monitored by MALDI-TOF MS after ZipTip cleanup and reaction times were extended as needed. Upon completion, cleaved products were purified by HPLC or solid-phase extraction using Bond Elut C18 (Agilent) cartridges.

### Mass spectrometry analysis
All MALDI-TOF MS analyses were carried out at the University of Illinois at Urbana-Champaign School of Chemical Sciences Mass Spectrometry Laboratory using either an Autoflex speed LRF MALDI-TOF (Bruker) or an UltrafleXtreme MALDI-TOF/TOF MS instrument (Bruker). All samples were either purified by HPLC or desalted using ZipTip C18 (Millipore) prior to analysis. Samples were co-spotted (1:1 v/v) on MALDI plates with a 50 mg/mL solution of matrix. The following matrices were used: 2,5-dihydroxybenzoic acid (Super-DHB, DHB, Sigma) in 80% MeCN:20% water supplemented with 0.1% TFA, sinapic acid (SA, Sigma) in 60% MeCN:40% water with 0.1% FA and a 2:1 (m:m) mixture of α-cyano-4-hydroxycinnamic acid (CHCA) and dihydroxybenzoic acid in 60% MeCN:40% water with 0.1% FA. Data was analyzed using flexAnalysis software (Bruker).

For high-resolution electrospray ionization (HR-ESI) mass spectrometry analyses, samples were purified by either HPLC or ZipTip, diluted 1:1 in 80% MeCN, 19% $H_2O$, 1% acetic acid and infused onto a ThermoFisher Scientific Orbitrap Fusion electrospray ionization (HR-ESI) mass spectrometer using an Advion TriVersa Nanomate 100. Mass spectrometry calibration was performed with Pierce LTQ Velos ESI Positive Ion Calibration Solution (ThermoFisher). The following parameters were used for MS(/MS) data acquisition: 100,000 resolution, 0.5–2 m/z isolation width (MS/MS), 35 normalized collision energy (MS/MS), 0.4 activation q value (MS/MS), and 30 ms activation time (MS/MS). Fragmentation was performed using collision-induced dissociation (CID) at 30%, 70% and 90%. Data analysis was conducted using the Qualbrowser application of Xcalibur software (ThermoFisher Scientific).

### Bacterial growth-suppression assays
Modified core peptides were purified by HPLC, lyophilized and re-dissolved in either 30% acetonitrile, 10% MeOH or 5% DMSO. The concentrations were then estimated using Pierce Quantitative Colorimetric Peptide Assay according to the manufacturer's protocol. Paper disc or agar well diffusion assays were used to evaluate the antimicrobial activity of the modified peptide cores against the panel of ESKAPE pathogens: *Enterococcus faecium* S235, *Staphylococcus aureus* NRS384, *Klebsiella pneumoniae* ATCC 27736, *Acinetobacter baumanii* ATCC 19606, *Pseudomonas aeruginosa* PAO1, and *Enterobacter cloacae* ATCC 29893. A starter culture of indicator strain was grown in LB medium under aerobic conditions at 37 °C for 16 h. Agar plates were prepared by combining 40 mL of molten Mueller Hinton (MH) agar (cooled to 42 °C) with 50 μL of dense culture (starting $OD_{600}$ -0.1–0.2). For *E. faecium*, 50% Brain Heart Infusion (BHI) broth and 50% MH was used. The seeded agar was poured into a sterile Petri dish and allowed to solidify at room temperature for 30 min after which 20 μL of modified peptides were spotted onto the agar plates. Plates were incubated at 37 °C for 15 h, and antimicrobial activity was qualitatively determined by the presence or absence of a zone of growth inhibition. For peptides that exhibited bioactivity, a serial dilution assay was performed to evaluate the minimum inhibitory concentrations (MIC) against the indicator strains. Stock solutions of modified core peptides were serially diluted in water and spotted onto a 96-well agar plate containing indicator strains at a starting $OD_{600}$ of 0.1–0.2 in MH. Plates were incubated at 37 °C for 16 h and MICs were determined by the presence or absence of a zone of growth inhibition for each dilution. The lowest concentration that inhibited visible microbial growth was reported as the MIC.

### Gas chromatography-mass spectrometry with a chiral stationary phase
To determine the stereochemistry of the lanthionine and methyllanthionine residues in the modified lanthipeptides, GC-MS analysis was carried out on hydrolysate of each peptide derivatized according to a previously published protocol[55]. Briefly, modified peptides were first purified on RP-HPLC and dissolved in 3 mL of 6 M DCl in $D_2O$ and sealed in a pressure tube. Each sample was heated to 110 °C in an oil bath for 24 h, cooled to room temperature, and the solvent removed by rotary evaporation. In a separate flask, a solution of methanolic HCl was prepared by adding 1.5 mL of acetyl chloride drop-wise to MeOH (5 mL) in an ice-water bath. About 3 mL of the MeOH-HCl was added to the hydrolysate residue and heated at 110 °C for 1 h under reflux. The reaction mixture was then allowed to cool to room temperature followed by the removal of the solvent by rotary evaporation. The resulting residue was resuspended in 3 mL of dichloromethane and cooled in an ice-water bath. Pentafluoropropionic anhydride (1 mL) was added to the reaction vessel, and the reaction mixture was heated to 110 °C for 1 h with reflux. The solution was cooled, and the solvent removed with a flow of nitrogen gas. The resulting residue was then dissolved in 0.2 mL of MeOH, centrifuged to remove any insoluble material, transferred to a clean vial, and stored at −20 °C prior to analysis. Gas chromatography−mass spectrometry (GC-MS) analysis was conducted on an Agilent HP 6890 N instrument equipped with a CP-Chirasil-L-Val (Agilent) column (25 m x 0.25 mm × 0.12 μm). Samples in MeOH (1 − 5 μL) were applied to the column via splitless injection (230 °C injector inlet temperature) with a helium flow at a rate of 1.7 − 2.0 mL/min. One of two temperature programs was used: (A) 5 min at 160 °C, followed by temperature increase of 3 °C/min to 190 °C, and then held for 6 min; or (B) held at 80 °C for 5 min, then raised to 200 °C at a rate of 4 °C/min, and then held at 200 °C for an additional 3 min. Selected-ion monitoring (SIM) mode was used to detect characteristic mass fragments for derivatized lanthionine or methyllanthionine (365 and 379 m/z, respectively). Stereochemical determination for derivatized samples was accomplished through co-injection with pure synthetic standards prepared according to reported methods[75].

Article

## NMR spectroscopy

Modified LanII-2A and LanII peptides were digested with LahT150 and endoproteinase AspN, respectively, HPLC-purified, and lyophilized. The core peptides were then re-dissolved in either 600 µL of 30% MeCN-*d3*, 10% MeOH-*d3*, or DMSO-*d6* (Cambridge Isotope Laboratories) depending on the solubility and transferred into an NMR tube. One dimensional $^1$H NMR, and two-dimensional homonuclear $^1$H-$^1$H-TOCSY (total correlation spectroscopy) and NOESY (nuclear Overhauser effect spectroscopy) spectra were acquired using a Varian INOVA 750 MHz spectrometer equipped with a 5 mm triple resonance ($^1$H-$^{13}$C-$^{15}$N) triaxial gradient probe with the use of VNMRJ 2.1B software and the BioPack suite of pulse sequences. The mixing time for TOCSY was 70 ms and that for NOESY was 300 ms. The optimal temperature for analysis was determined to be 25 °C based on peak width. The amide protons were well dispersed in the LanII sample, but severe peak broadening was observed for LanII-2A in all solvents in which the peptide was soluble. For LanII-2A, the TOCSY and NOESY spectra were used for sequential assignment of all spin systems. The $^1$H-$^{13}$C HSQC (heteronuclear single quantum correlation) and $^1$H-$^{13}$C HMBC (heteronuclear multiple-quantum correlation) spectra for LanII were collected on a Bruker Avance NEO 600 MHz spectrometer with a 5-mm prodigy BBO probe using Topspin 4.1.4 software. Both spectra were processed in MestReNova 11.0.3. These two spectra were used to confirm the thioether linkages in addition to the NOESY data. chemical shift assignments based on $^1$H-$^1$H-TOCSY and $^1$H-$^{13}$C-HSQC data are shown in Supplementary Tables 2 and 3 respectively.

## Reporting summary

Further information on research design is available in the Nature Research Reporting Summary linked to this article.

## Data availability

We declare that all the data supporting the findings of this study are available within the main text, Supplementary data 1 and 2, and Supplementary information documents. Gene sequences used in this study were obtained from NCBI: https://www.ncbi.nlm.nih.gov/; NCBI accessions for each gene can be found in Supplementary data 1. Additional NCBI accession used: WP_036102482.1. The DNA sequences of 96 plasmids used in this study are provided as Supplementary data 2. All plasmids and raw data are available upon request from the corresponding authors. Source data are provided with this paper.

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

## Acknowledgements

We thank P. Yau and J. Arrington in the Roy J. Carver Biotechnology Center at the University of Illinois at Urbana-Champaign for assistance with the MS/MS analysis. This work was supported in part by a grant from the National Institutes of Health (AI144967 to D.A.M., W.A.vdD., and H.Z.) and fellowships GM140621 (to R.S.A.), T32-GM070421 (to A.J.B. and S.M.E.), Francis M. and Harley M. Clark Microbiology Fellowship (to S.R.), T32-GM136629 and DGE 21-46756 (to A.J.R). W.A.vdD is an Investigator of the Howard Hughes Medical Institute. BioRender (biorender.com) was used to create Fig. 2.

## Author contributions

All authors designed experiments, analyzed data, and assisted in the writing and editorial process. R.S.A., C.S., A.J.B., S.M.E., S.R., M.A.S., I.R.B. H.L., A.J.R., H.R., Q.T., L.A.H., R.S., L.Z., A.M.F., and T.W.P. performed the experiments. D.A.M., W.A.vdD., and H.Z. conceived and supervised the overall project.

## Competing interests

D.A.M. is a co-founder and owns stock options in Lassogen, Inc. D.A.M., W.A.vdD., and H.Z. declare that a provisional patent application based on the methods described within has been filed with the University of Illinois Office of Technology Management (US63/354,994 (2022)). The remaining authors declare no other competing interests.
