## [Peer Review File · Nature Communications]

A Scalable Platform to Discover Antimicrobials of Ribosomal OriginREVIEWER COMMENTS

Reviewer #1 (Remarks to the Author):

The authors generated an automated high-throughput-platform to generate RiPPs. The process starts from their bioinformatics platform, which identifies RiPPs of interest and is then transformed into the heterologous expression in E.coli, followed by MS-analysis. Hits were then evaluated for their antibiotic activity and in one case the structure was secured by NMR analysis.

Exemplary biosynthetic gene cluster of each RiPP classes and different degrees of complexity were chosen from taxonomically different organisms.

Even if some of the experiments did not work, the amount of work and the success rate at the level of heterologous expression is impressive and so is the number of newly discovered RiPPs. Also, that the authors could demonstrate activity against an ESKAPE pathogen raises even more the significance of their findings.

The applied methods are absolutely appropriate; the analytics as I would expect – the majority can be done with MS but at least one structure is proven by NMR.

Scientifically and technically this study is a blast ! Congratulations !

Nobody has done that on such a scale (not automated anyway) and such a success rate.

It can be published as it is (I usually never say that) – they only have to take care about their own comment on page 19 (re-calculate and fill in the correct numbers).

Reviewer #2 (Remarks to the Author):

In their manuscript entitled "A Scalable Platform to Discover Antimicrobials of Ribosomal Origin" Ayikpoe et al. present a new platform for the production of uncharacterised ribosomally synthesized and post-translationally modified peptides (RiPPs). First, the authors use the bioinformatic RiPP discovery tool RODEO, developed by the Mitchell group, to identify and select target biosynthetic gene clusters. I particularly liked the overall spread, both in terms of organisms and RiPP classes covered. An initial selection of 96 BGCs is then selected for automatic refactoring and expression in E. coli. It seems to be an advantage of RiPPs that many of them can be produced in E.coli without too much trouble, and the authors managed to obtain 27 "novel" compounds. The quotation marks are only there because the authors were unfortunate that some of the compounds were reported by other groups before this publication, but that should not detract from this impressive manuscript. Selected compounds are taken all the way through to biological activity testing, with a focus on high-priority pathogens. The authors take great care to characterise the compounds, with detailed characterisation reserved for the high-priority (bioactive) examples. Overall, this manuscript is a very enjoyable and impressive read, and I wholeheartedly support publication after the following very minor issues have been addressed:

The authors do not explain the ESKAPE pathogens until the results section (line 189) where they list which they tested against, however they are mentioned before this in the introduction section (line 77). I think it would make sense to establish what the ESKAPE pathogens are and why they are such an important target in the introduction.

Figure 1 – The writing is quite small in comparison to Fig 2 etc and should maybe be made a bit bigger. The authors also abbreviate LP and CP in this figure legend but never use these abbreviations again and write out leader peptide and core peptide every time.

Line 497 – start of sentence should have capital letter "Doubling"

I know that there is a list of the 27 modified peptides produced, but maybe it would be nice to have a final figure showing the three new RiPP product that show antimicrobial properties against the

ESKAPE pathogens – Maybe showing the plates/ control plates or something alongside the structure of each so there is more primary data included. The authors have some plates shown in the supplementary, but it could be nice to show something like this in the main paper?

Supplementary – In the Supp Fig 1 and 2, a) b) etc are in bold but from Fig 6 onwards they aren't – should they all be in bold or not?

Reviewer #3 (Remarks to the Author):

The authors are taking advantage of the inherent capability of an automated, high-throughput DNA and organism engineering platform to interrogate the vast amounts of data that are now coming out of automated data-mining tools. Techniques like the authors have employed in using a biofoundry are exactly what these platforms have been designed to accomplish.

Establishing a high-throughput pipeline to make a first pass at a collection of BCGs with the aim of screening for anti-microbials is an interesting approach. I would be interested to see in future if/when this high-throughput pipeline is used, based on the data gathered in this and other studies, to group the BCGs and refactor for expression into a couple of other host organisms to explore the benefits of other host systems for expression of intractable/silent compounds.

The work is significant in the field of high-throughput bio-engineering as well as anti-microbial compound discovery. It also demonstrates a good use of pairing in-silico discovery with in-vitro verification.

Commenting specifically on the biofoundry work, the work meets expected standards and the methodology is sound.

There is enough detail provided for the work to be reproduced.

Janet Reid

Reviewer #1:

1. It can be published as it is (I usually never say that) – they only have to take care about their own comment on page 19 (re-calculate and fill in the correct numbers).

Response: We filled in the correct number in the revised manuscript. We thank this reviewer for the supportive comments.

Reviewer #2:

1. The authors do not explain the ESKAPE pathogens until the results section (line 189) where they list which they tested against, however they are mentioned before this in the introduction section (line 77). I think it would make sense to establish what the ESKAPE pathogens are and why they are such an important target in the introduction.

Response: We agree and have added two sentences to the first paragraph of the paper discussing the ESKAPE pathogens.

2. Figure 1 – The writing is quite small in comparison to Fig 2 etc and should maybe be made a bit bigger. The authors also abbreviate LP and CP in this figure legend but never use these abbreviations again and write out leader peptide and core peptide every time.

Response: We thank the reviewer for these suggestions. The text in Figure 1 has been changed from size 8 to size 12. The LP and CP abbreviations have been deleted.

3. Line 497 – start of sentence should have capital letter “Doubling ...”

Response: The period that was before this word was corrected to a comma.

4. I know that there is a list of the 27 modified peptides produced, but maybe it would be nice to have a final figure showing the three new RiPP product that show antimicrobial properties against the ESKAPE pathogens – Maybe showing the plates/ control plates or something alongside the structure of each so there is more primary data included. The authors have some plates shown in the supplementary, but it could be nice to show something like this in the main paper?

Response:

We acknowledge the importance of obtaining the final structures for all three bioactive compounds. However, LanII-2A showed poor solubility and peak broadening in NMR compatible solvents. Thus, the ring connectivity could not be determined by NMR at this time. We are exploring alternative means to obtain structural information but

believe that the most important contribution of this work is the production platform that can be implemented in other labs. LanII-2B is only partially modified and hence determining the current structure would not be that of the natural product. We prefer to first explore other routes that may give the final structure before we embark on the substantial task of obtaining sufficient material for structure determination.

We agree that it would be valuable to provide data that show how we determined the MIC. We added a new panel to supplementary figure 52. We feel that the pictures of the plates are not good images for the main text which already has many figures and therefore we left them in the Supplementary Information.

5. Supplementary – In the Supp Fig 1 and 2, a) b) etc are in bold but from Fig 6 onwards they aren't – should they all be in bold or not?

Response: Thank you for pointing that out. These panel letterings in the figure captions have been bolded.

Reviewer #3:

Response: We thank this reviewer for the supportive comments.

REVIEWER COMMENTS

Reviewer #2 (Remarks to the Author):

I would like to thank the authors for implementing all changes that were requested and for their explanation why one suggestion could not be addressed at this time. I am happy for the manuscript to be published as is.

Reviewer #4 (Remarks to the Author):

The authors report a scalable platform that combines high-throughput bioinformatics with automated biosynthetic gene cluster (BGC) refactoring for rapid evaluation of uncharacterized RiPP BGCs. This platform takes advantage of a previous biofoundry, the iBioFAB.

This is an ambitious work and a great effort that represents a major advance in the discovery of new RiPPs. The ratio of success in refactoring RiPP BGCs is very high and the platform could be implemented in other labs or service them. All in all, a really good article.

I believe, however, that the application of this platform to the discovery of new antimicrobials still need important optimizations. A significant number of the newly discovered RiPPs were isolated as only partially modified products and therefore, antibacterial activities may have been missed.

It is also challenging to target Gram-negative ESKAPE pathogens with RiPPs, so it is interesting that a class-II lanthipeptide (LanII-2A) showed low μM activity against *K. pneumoniae*.

Among the new modified peptides discovered, the LanII BGC product is claimed to have a unique ring pattern amongst known lanthipeptides. The authors also found orthologs of the precursor peptide in several genera of Rhodanobacteraceae and they propose the name rhodanocins for this new subfamily.

My main concerns relate to the structural characterization of LanII. The authors rely on NEM/DTT derivatizations, HRMS-MS and ^1H $1\text{D}/2\text{D}$ NMR (^1H , TOCSY, NOESY). Although the proposed structure and ring pattern appear reasonable, I believe that the data provided may not be entirely sufficient for an unambiguous structure determination:

i) The presence in AspN-digested LanII of the encoded amino acids (modified or not) in the peptide core (except the N-terminal Thr, as expected) was confirmed in the TOCSY spectrum. Next, it appears that amino acid connectivity was established by NOESY correlations between NH amide (i) and H_α protons (i-1), although the sentence referring to this (lines 443-444 of the revised manuscript) is not fully clear. Considering that this connectivity was confirmed based only on nOe but not on HMBC correlations, it would be nice to include expansions of the NOESY spectrum showing the corresponding cross-peaks.

ii) The ring pattern in LanII (A,B,C) was established by NOESY correlations between the β methylene protons (also H_α proton for C7) across the thioether linkages. Although this approach has been used in several works, such correlations may result from the three-dimensional structure of the lanthipeptide, which may place two separate (Me)Lan residues in close proximity (especially for peptides with such globular structures). ^1H - ^{13}C HMBC experiments would be very helpful to confirm the sequence connectivity and, above all, to unambiguously determine the ring pattern. If the amount of LanII isolated and purified is sufficient, HSQC and HMBC experiments should be performed. Otherwise, the authors should try to confirm the ring pattern by chemical modifications (partial desulfurizations with nickel boride under deuterated conditions should work) and subsequent tandem mass spectrometry.

Considering that LanII is claimed to belong to a new subfamily for which a significant number of orthologs have been found, and to avoid possible erroneous structural assumptions in future members of this family, I believe it would be worthwhile to do this experimental effort.

Apart from the above, a few other minor comments:

1) Line 212: To better contextualize for a broad readership, the authors should specify that a fifth class of lanthipeptides has been described to date.

2) Lines 236-237: The sentence..“The peptides from LanII-2A and LanII-2C were dehydrated three and five times, respectively, and fully cyclized” ...is true for LanII-2C but not for LanII-2A. According to supplementary figure 7, LanII-2A effectively dehydrated 3 times but cyclized only 2 times.

3) Figure 5a / Figure S5: The PTMs labeled in purple for LanII are not the same as those labeled in supplementary Figure 5.

In addition, Supplementary Figure 5 is confusing, as LanII has 4 dehydrations but only 3 cyclizations, as shown by the addition of 1 DTT. However, the exact mass depicted in the lower panel of Figure S5 is the same as the upper panels and the peaks are aligned with each other. It appears that the X-axis scale is shifted and matches the expected mass value for the addition of 1 DTT.

4) Figure 5a / Figure S7: The PTMs labeled in purple for LanII-2A are not the same as those labeled in supplementary Figure 7.

5) The legends at the bottom of the supplementary figures need to be revised. From Figure S22 onwards, they are numbered starting again from S1, so many numbers are repeated (the numbers in the SI table of contents appear to be correct).

6) In the supplementary figure showing the TOCSY spectrum of LanII (according to the lower legend it would be supplementary figure 41b, but I think it is wrong), where it says... "Abu17", it should read... "Dhb17" and where it says.. "Met3", it should read... "Met2".

Also, an expansion showing the spin system H α /H β of Asp1 should be included

7) In the supplementary table 2, NMR data of Ala23 residue are missing. Its spin system should be shown as well as a TOCSY expansion

Reviewer #4:

The authors report a scalable platform that combines high-throughput bioinformatics with automated biosynthetic gene cluster (BGC) refactoring for rapid evaluation of uncharacterized RiPP BGCs. This platform takes advantage of a previous biofoundry, the iBioFAB.

This is an ambitious work and a great effort that represents a major advance in the discovery of new RiPPs. The ratio of success in refactoring RiPP BGCs is very high and the platform could be implemented in other labs or service them. All in all, a really good article.

Response: We thank this reviewer for the supportive comments and for the critical read that allowed us to improve the manuscript.

I believe, however, that the application of this platform to the discovery of new antimicrobials still need important optimizations. A significant number of the newly discovered RiPPs were isolated as only partially modified products and therefore, antibacterial activities may have been missed.

It is also challenging to target Gram-negative ESKAPE pathogens with RiPPs, so it is interesting that a class-II lanthipeptide (LanII-2A) showed low μM activity against *K. pneumoniae*.

Among the new modified peptides discovered, the LanII BGC product is claimed to have a unique ring pattern amongst known lanthipeptides. The authors also found orthologs of the precursor peptide in several genera of Rhodanobacteraceae and they propose the name rhodanocins for this new subfamily.

My main concerns relate to the structural characterization of LanII. The authors rely on NEM/DTT derivatizations, HRMS-MS and ^1H 1D/2D NMR (^1H , TOCSY, NOESY). Although the proposed structure and ring pattern appear reasonable, I believe that the data provided may not be entirely sufficient for an unambiguous structure determination:

i) The presence in AspN-digested LanII of the encoded amino acids (modified or not) in the peptide core (except the N-terminal Thr, as expected) was confirmed in the TOCSY spectrum. Next, it appears that amino acid connectivity was established by NOESY correlations between NH amide (i) and $\text{H}\alpha$ protons (i-1), although the sentence referring to this (lines 443-444 of the revised manuscript) is not fully clear. Considering that this connectivity was confirmed based only on nOe but not on HMBC correlations, it would be nice to include expansions of the NOESY spectrum showing the corresponding cross-peaks.

Response: In the newly revised manuscript we do both, by including newly acquired HMBC and HSQC data and providing expansions of the NOESY spectrum.

ii) The ring pattern in LanII (A,B,C) was established by NOESY correlations between the β methylene protons (also $\text{H}\alpha$ proton for C7) across the thioether linkages. Although this approach has been used in several works, such correlations may result from the three-dimensional structure of the lanthipeptide, which may place two separate (Me)Lan residues in close proximity (especially for peptides with such globular structures). ^1H - ^{13}C HMBC experiments would be

very helpful to confirm the sequence connectivity and, above all, to unambiguously determine the ring pattern. If the amount of LanII isolated and purified is sufficient, HSQC and HMBC experiments should be performed. Otherwise, the authors should try to confirm the ring pattern by chemical modifications (partial desulfurizations with nickel boride under deuterated conditions should work) and subsequent tandem mass spectrometry.

Response: The original amount of LanII that we obtained was not sufficient, but we have now scaled up the production in *E. coli* and although the amount of material is still low, we managed to obtain HSQC and HMBC data that agree with our original assignments. These data have been added in a new figure (Supp. Fig. 55) and the data in a new Supp. Table 3.

Considering that LanII is claimed to belong to a new subfamily for which a significant number of orthologs have been found, and to avoid possible erroneous structural assumptions in future members of this family, I believe it would be worthwhile to do this experimental effort.

Response: We have performed HSQC and HMBC experiments to confirm the proposed ring patterns for LanII (see new Supplementary Figure 55 and new Supp. Table 3). The HSQC and HMBC data support the NOESY data. We also included a few sentences in the structure determination section of the results and methods sections.

Apart from the above, a few other minor comments:

1) Line 212: To better contextualize for a broad readership, the authors should specify that a fifth class of lanthipeptides has been described to date.

Response: We rephrased the text to mention the five classes of lanthipeptides.

2) Lines 236-237: The sentence..“The peptides from LanII-2A and LanII-2C were dehydrated three and five times, respectively, and fully cyclized” ...is true for LanII-2C but not for LanII-2A. According to supplementary figure 7, LanII-2A effectively dehydrated 3 times but cyclized only 2 times.

Response: LanII-2A contains two Cys residues and LanII-2C contains five Cys residues all of which are involved in the formation of two and five lanthionine rings respectively. That is why we used the term fully cyclized. Although LanII-2A is indeed dehydrated three times, it can only cyclize two times.

3) Figure 5a / Figure S5: The PTMs labeled in purple for LanII are not the same as those labeled in supplementary Figure 5.

Response: We agree and thank the reviewer for pointing this out. A Ser should have been colored in purple in Fig 5a. The PTMs in LanII in Figures 5a/S5 have been re-colored to be consistent with the data.

In addition, Supplementary Figure 5 is confusing, as LanII has 4 dehydrations but only 3 cyclizations, as shown by the addition of 1 DTT. However, the exact mass depicted in the lower

panel of Figure S5 is the same as the upper panels and the peaks are aligned with each other. It appears that the X-axis scale is shifted and matches the expected mass value for the addition of 1 DTT.

Response: LanII is dehydrated four times and cyclized three times (based on both lack of NEM adducts and NMR data). Actually, no mass change was observed after treatment of the modified peptide with DTT and this was further confirmed by HRMS and orbitrap data. Hence, we do not see addition of 1 DTT (the X-axis scale was not shifted and no DTT adduct was observed). The possible reasons for this lack of DTT adduct formation may be that the fourth dehydroamino acid is embedded in a ring and no longer exposed for reactivity upon lanthionine ring formation, that the dehydro amino acid is not reactive because of conformational restraints (which may force the two π systems of the dehydro amino acids out of conjugation), or that the non-cyclized residue is a Dhb which is much less reactive than Dha towards DTT (or a combination of these three reasons). The presence of a free dehydrobutyrine in the LanII structure was confirmed by the presence of a quartet at 6.80 ppm in the ^1H NMR spectrum, and the TOCSY and HSQC data (Supplementary Figure 54 and Supplementary Tables 2 and 3). This Dhb is indeed inside a ring, which may have prevented its reaction with DTT.

The reviewer is correct that the masses listed in the previous Supp. Figure 5 were not in agreement with the X-axis. Since we generated more of LanII for the NMR data that the reviewer requested, we used the new samples to repeat the DTT and NEM assays. As expected, no adducts were observed and the mass of the peptide is this time as expected. We replaced the old figure with the new data.

4) Figure 5a / Figure S7: The PTMs labeled in purple for LanII-2A are not the same as those labeled in supplementary Figure 7.

Response: Indeed an additional Thr should have been colored purple in Fig 5a. The PTMs in LanII-2a Figures 5b/S7 have been re-colored to be consistent with the data.

5) The legends at the bottom of the supplementary figures need to be revised. From Figure S22 onwards, they are numbered starting again from S1, so many numbers are repeated (the numbers in the SI table of contents appear to be correct).

Response: Thank you for pointing this out. The legends have been revised.

6) In the supplementary figure showing the TOCSY spectrum of LanII (according to the lower legend it would be supplementary figure 41b, but I think it is wrong), where it says... "Abu17", it should read... "Dhb17" and where it says... "Met3", it should read... "Met2". Also, an expansion showing the spin system $\text{H}\alpha/\text{H}\beta$ of Asp1 should be included

Response: Abu should indeed be Dhb in Supp Figure 54 and we made the change to the figure. Met3 is actually correct and is based on the numbering of the core peptide (not the AspN-digestion), but our drawing indeed confused the general numbering of the peptide and we have clarified the drawing and renumbered the residues in the current revised figure. The TOCSY

spectrum has also been expanded to include the spin systems H α /H β of Asp2 (Asp1 in the prior numbering).

7) In the supplementary table 2, NMR data of Ala23 residue are missing. Its spin system should be shown as well as a TOCSY expansion

Response: Assignment for the C-terminal Ala23 (now Ala24 with the updated numbering as mentioned in response to the previous point) has been included in the TOCSY spectrum and the data are now listed in Supplementary Table 2.

REVIEWERS' COMMENTS

Reviewer #4 (Remarks to the Author):

I would like to thank all the authors for their efforts to respond to all the questions raised.

I consider that all issues have been satisfactorily addressed and believe that the manuscript can be published without changes.